behaviour/environmental science

*Lepomis gibbosus*, *Pseudorasbora parva*, invasive non-native species, flow velocity, turbulence, fish dispersal

**Author for correspondence:**
V. Muhawenimana
e-mail: muhawenimanav@cardiff.ac.uk

# Temperature surpasses the effects of velocity and turbulence on swimming performance of two invasive non-native fish species

V. Muhawenimana[1], J. R. Thomas[2], C. A. M. E. Wilson[1], J. Nefjodova[2], A. C. Chapman[2], F. C. Williams[3], D. G. Davies[3], S. W. Griffiths[2] and J. Cable[2]

[1]School of Engineering, Cardiff University, Cardiff CF10 3AA, UK
[2]School of Biosciences, Cardiff University, Cardiff CF10 3AX, UK
[3]National Fisheries Services, Environment Agency, Huntingdon PE28 4NE, UK

VM, 0000-0002-9538-2229; JC, 0000-0002-8510-7055

Global climate change continues to impact fish habitat quality and biodiversity, especially in regard to the dynamics of invasive non-native species. Using individual aquaria and an open channel flume, this study evaluated the effects of water temperature, flow velocity and turbulence interactions on swimming performance of two lentic, invasive non-native fish in the UK, pumpkinseed (*Lepomis gibbosus*) and topmouth gudgeon (*Pseudorasbora parva*). Burst and sustained swimming tests were conducted at 15, 20 and 25°C. Acoustic Doppler velocimetry was used to measure the flume hydrodynamic flow characteristics. Both *L. gibbosus* and *P. parva* occupied the near-bed regions of the flume, conserving energy and seeking refuge in the low mean velocities flow areas despite the relatively elevated turbulent fluctuations, a behaviour which depended on temperature. Burst swimming performance and sustained swimming increased by up to 53% as temperature increased from 15 to 20°C and 71% between 15 and 25°C. Furthermore, fish test area occupancy was dependent on thermal conditions, as well as on time-averaged velocities and turbulent fluctuations. This study suggests that invasive species can benefit from the raised temperatures predicted under climate change forecasts by improving swimming performance in flowing water potentially facilitating their further dispersal and subsequent establishment in lotic environments.

# 1. Introduction

Human alterations to freshwater ecosystems through hydro-engineering and water management activities, combined with global climate change, continue to alter the aquatic environment. Disruption of the natural environment has resulted in the fragmentation of river habitats and directed research into understanding fish behaviour in altered flows, with the ultimate aim of restoring eco-service functions [1–3]. Climate change, on the other hand, impacts all environments, significantly affecting biodiversity and species distribution [4,5]. Extreme hydrological events and increasing air and water temperatures also affect the dispersal and establishment of invasive species [6–8]. The management of invasive species, which are currently introduced and propagated by socio-economic drivers [8], further strain freshwater ecosystems and add to the challenge of management efforts to minimize the impact of hydraulic engineering [3,9–11].

Quantification of fish swimming behaviour relative to flow properties is critical for the design of fish management and passage solutions in water engineering applications. Fish swimming performance is highly dependent on environmental factors, particularly water temperature, which can drastically alter fish physiology [12,13]. The burst, sustained and critical swimming velocities of fish are used as indicators of ability to navigate natural or altered flows for short and long distance movements and, therefore, are used to link fish swimming behaviour with altered flow hydrodynamics [14–18]. Since temperature directly affects fish metabolism [19], some species benefit from temperature increases, which improves their swimming velocities, while others respond negatively [12]. This key effect of thermal regimes on swimming performance is likely to intensify due to global warming, which will not only change storm frequency, timing and duration, and the resulting river runoff regimes, but most importantly water quality and temperature, with direct impacts on aquatic ecosystems [4,20–22].

Thermal alterations to rivers and other fresh waters will also influence the diversity of fish that thrive in these habitats. As a result of anthropogenic activity, invasive non-native fish species are being increasingly introduced to freshwater systems worldwide, either as deliberate introductions for aquaculture, ornamental purposes and fishery enhancement stockings [23] or accidentally as 'contaminants' of permitted introductions [24]. A small proportion (approx. 1%) of these introductions have led to non-native fishes establishing invasive populations, causing ecological and/or economic damage to the receiving environment and ecosystems [8]. Thus, successful invaders must pass through a series of stages, including transport, introduction, establishment and spread [8]. For fish, thermal changes in rivers will potentially have a drastic impact on their ability to swim, disperse and spread [7,10,20]. In the field of ecohydraulics, studies of fish swimming behaviour in altered and turbulent flows have largely overlooked the effects of thermal regime variations. The few studies evaluating changes in fish swimming performance under different thermal regimes in relation to flow velocity and turbulence are inconclusive; finding either increased swimming costs due to temperature and flow velocity increases [25] or high variability of temperature effects [26,27].

The present study focused on two invasive non-native UK freshwater fish species, the pumpkinseed (*Lepomis gibbosus*) and the topmouth gudgeon (*Pseudorasbora parva*). *Lepomis gibbosus* is a sunfish native to North America but widely distributed throughout Europe, with limited populations in the UK [28]. *Pseudorasbora parva* is native to East Asia but has achieved pan-continental distribution across Europe [29], and in England and Wales is subject to an ongoing eradication programme [9]. *Lepomis gibbosus* are considered less invasive in the UK than in Europe due to current climatic conditions [30]; additionally, both *L. gibbosus* and *P. parva* are primarily constrained to lentic environments. However, climate change could facilitate their dispersal from lentic to lotic environments [7,31]. Despite this, the effects of water temperature on the swimming performance of both species are unknown. Here, we tested whether temperature affects burst and sustained swimming performance of both *L. gibbosus* and *P. parva* from non-native English populations, in addition to evaluating the combined effect of temperature, velocity and turbulence on occupancy of the test area zones and sustained swimming performance.

# 2. Methods

## 2.1. Animal origin and maintenance

*Lepomis gibbosus* (*n* = 133) were sourced from a small fishery pond in Southern England (51°21′43″ N, 2°49′18″ W) in September 2015 by seine net (25 × 2.5 m) and immediately transported to the aquarium facilities at Cardiff University. *Pseudorasbora parva* (*n* = 105) were captured from a pond in southwest

**Table 1.** Number of fish N, standard length SL (mm) (mean ± s.d.) of *L. gibbosus* and *P. parva* fish tested for burst and sustained swimming performance at three temperatures following four weeks acclimatization to these conditions.

| fish species | test | temperature (°C) | N | SL (mm) |
|---|---|---|---|---|
| *Lepomis gibbosus* | burst | 15 | 15 | 64.7 (±2.6) |
| | | 20 | 15 | 64.4 (±2.4) |
| | | 25 | 15 | 62.3 (±1.9) |
| | sustained | 15 | 10 | 69.4 (±3.2) |
| | | 20 | 13 | 63.8 (±2.5) |
| | | 25 | 9 | 61.4 (±2.1) |
| *Pseudorasbora parva* | burst | 15 | 10 | 56.9 (±2.4) |
| | | 20 | 10 | 52.7 (±1.6) |
| | | 25 | 9 | 54.8 (±2.0) |
| | sustained | 15 | 15 | 51.9 (±1.4) |
| | | 20 | 9 | 51.4 (±3.5) |
| | | 25 | 12 | 53.2 (±1.9) |

England in July 2016 using circular fish traps, comprising a circle alloy frame covered in 2 mm mesh baited with fishmeal pellets (21 mm diameter), and then again immediately transported to Cardiff. Each species was maintained in multiple aquaria, 30 fish per 100 l, filled with dechlorinated water and provided with plant pot refugia. The fish were maintained at 15°C under a 12 h : 12 h light : dark regime, fed daily with frozen Tubifex bloodworm and supplemented with live *Daphnia* weekly.

Approximately two weeks following transport to Cardiff University School of Biosciences Aquaria, all fish were anaesthetized using MS-222, measured (standard length, measured to the nearest mm), weighed (in g, to two decimal places) and a photograph taken of each fish using an iPhone 6S to allow the body and caudal fin area of each fish to be calculated using ImageJ [32]. Juvenile *L. gibbosus* could not be sexed morphologically, and dissection and examination of gonads confirmed that they were immature. *Pseudorasbora parva* were sexed following dissection at the end of the study. Additionally, *L. gibbosus* were PIT tagged (passive integrated transponder; 7 × 1.35 mm; ISO 11784 (134.2 kHz); Loligo® Systems, Toldboden, Denmark) for individual identification. All PIT tags were retained, and no mortality occurred as a result of tagging. *Pseudorasbora parva* reportedly suffer high mortality and PIT tag rejection [33] and, therefore, were not tagged. Following a fortnight recovery period after PIT tag insertion (*L. gibbosus*) or sham-handling (*P. parva*), fish were separated into three groups (six tanks per species) and acclimatized at 15, 20 or 25°C for one month (table 1) prior to experimental testing at the same temperature. Fish were fasted for 12 h prior to testing. All experiments for *L. gibbosus* were completed during November 2015, while those for *P. parva* were completed during September 2016 between 8.00 and 18.00 h.

## 2.2. Burst swimming

Burst swimming trials within individual aquaria were performed on *L. gibbosus* and *P. parva* four weeks after acclimatization to 15, 20 or 25°C in the School of Biosciences. *Lepomis gibbosus* and *P. parva* are known to exhibit a two-part burst swimming response, characterized by contraction of the body into a C-shape [34,35] followed by an expansion of the body generating thrust. Individual fish, tested once each, were placed in shallow water (10 cm) in the inner arena (21.6 × 21.6 × 12 cm) (figure 1), and this water was maintained at either 15, 20 or 25°C with a standard aquarium heater. After a 30 min acclimatization period in the arena, the fish were exposed to a stimulus (dropping a golf ball) every 5 min for 15 min (three golf balls per trial). We calculated the speed of the initial C-start responses to each stimulus following a previous study [36], which we averaged to obtain the burst swimming speed for each trial. All golf balls were dropped remotely outside the swimming arena from the same height above a covered side of the outer arena. The arena was lit using a fluorescent strip-light. The fish startle response was recorded using a digital camera (Sony HDR-CX405) with 1440 × 1080 resolution at 60 frames s$^{-1}$, and the burst swimming speed (cm s$^{-1}$) calculated using 'Tracker' software [37].

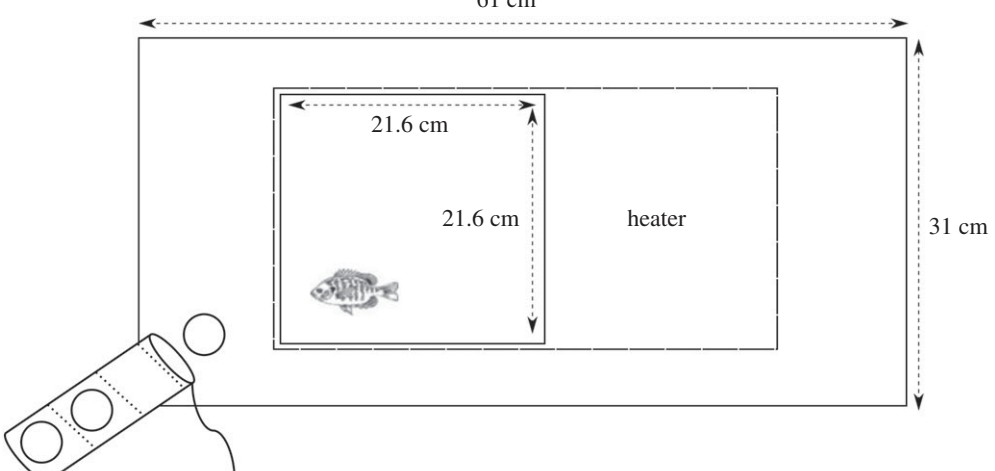

**Figure 1.** Experimental arena for testing the startle response (burst swimming), where *P. parva* and *L. gibbosus* were tested in the 21.6 × 21.6 × 10 cm inner area. Diagram not to scale.

## 2.3. Sustained swimming

Fish sustained swimming was assessed in a recirculating open channel flume (length: 10 m; width: 0.3 m and depth: 0.3 m) in the Hydro-environmental Research Centre, School of Engineering, Cardiff University. The flume (Armfield) consists of an upstream stilling basin, a 10 m long open channel with a stainless steel bed and glass walls, and a downstream reservoir. The flume had a slope of 1/1000. Flow was recirculated using a pump with $30 \, l \, s^{-1}$ discharge capacity, and the discharge was measured using an electromagnetic flowmeter ($\pm 0.3\% \, l \, s^{-1}$). The water was cooled to $15 \pm 1°C$ using a DD DC chiller or heated to 20 and $25 \pm 1°C$ using an Electro Titanium Digital heater. Sustained swimming was measured by recording the time to fatigue, $T_f$ (min) of fish by exposing fish to stepwise flow velocity increases ($5 \, cm \, s^{-1}$) in 10 min increments [35]. Each fish underwent the sustained swimming test once. The flow depth was kept constant at 13.5 cm using a tailgate weir on the downstream end of the flume and velocity increments were made by increasing the discharge by $2 \, l \, s^{-1}$, equivalent to velocity increments of $5 \, cm \, s^{-1}$ [35]. The time to fatigue test covered a velocity range of 9.28 to $53.83 \, cm \, s^{-1}$, as shown in table 2. The test section was 1.21 m long, delimited by honeycomb flow diffusers positioned 3.66 m upstream and 4.87 m downstream from the flume inlet.

Fish were introduced in the downstream end of the test section, and acclimatized to the flume for 30 min at the lowest available discharge of $1.76 \, l \, s^{-1}$, which corresponds to a cross-sectional averaged velocity ($U_0$) of $4.34 \, cm \, s^{-1}$. The swimming tests were recorded using a MacBook Air laptop, positioned to view the fish through the flume glass walls. Videos were analysed to quantify fish behaviour using JWatcher (v. 1.0), by recording their longitudinal, lateral and vertical position in the test section throughout the swimming test. The flume test area was divided into subsections along the centre and near the walls, as well as elevations in the water column, according to where fish swam. Time to fatigue, $T_f$ (min), which was the duration fish were able to swim continuously throughout the sustained velocity swimming test before exhaustion, i.e. when the fish touched the downstream honeycomb flow straightener, is used here as a cumulative time indicator of fish swimming performance. $T_f$ can be obtained as: $T_f = n^* t_{ii} + t_i$, where $n$ is the number of velocity steps fully completed, $t_i$ is the time fish swam at the fatigue velocity and $t_{ii}$ is the step duration.

## 2.4. Open channel velocity data collection and post-processing

Velocity data were collected using a downward looking Nortek Acoustic Doppler Velocimeter (ADV) at 200 Hz sampling frequency and 300 s sampling time. In the longitudinal direction ($x$), three measurement positions were located at 3.69, 4.26 and 4.86 m (A, B and C locations) from the flume inlet (figure 2). At these longitudinal locations, measurements were made along the flume centreline and within 25 mm of the flume wall (D, E and F locations), marking two sampling positions in the transverse direction ($y$). In the vertical direction, six points, 10 mm apart were measured at each location. This made six profiles with six points per Reynolds number. Due to the configuration of the ADV, only a portion of the flow depth

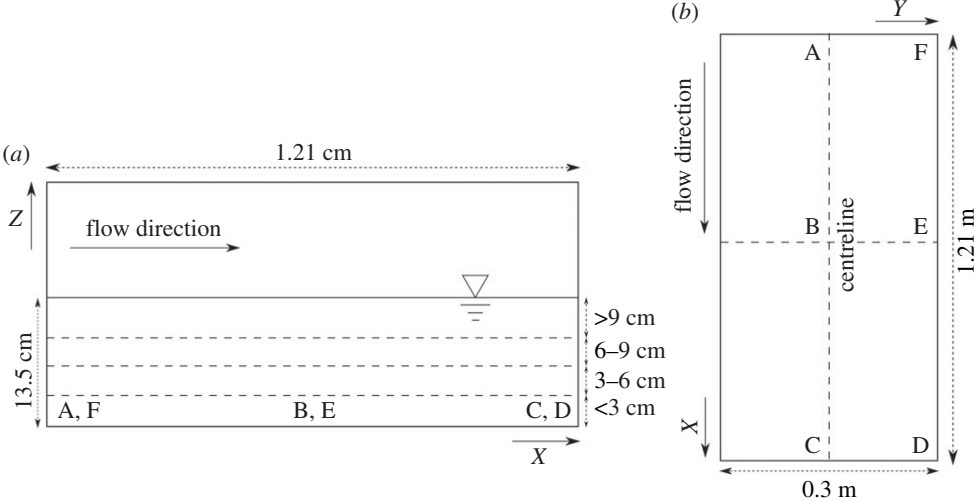

**Figure 2.** A schematic diagram showing the (*a*) side view and (*b*) aerial view of the test section of the recirculating open channel flume (not to scale, 1.21 m long with a 13.5 cm flow depth) for fish behaviour and velocity measurements. Locations at which the velocity profiles were measured along the flume centreline (A–C) and within 25 mm of the flume wall (D–F) are indicated with dashed lines, together with subdivisions of the water column used in the analysis of habitat occupancy. The underlined triangle in (*a*) indicates the water surface.

**Table 2.** Flume flow velocity and discharge values for velocity measurements at 15°C, and during sustained velocity swimming tests in *L. gibbosus* and *P. parva* over 120 min. $T_s$ is the start time and $T_e$ is the end time of each velocity step. $Q$ is the volumetric discharge, $U_0$ is the cross-sectional average velocity and $Re$ is the Reynolds number (where $Re = U_0 R_0/\nu$, the water kinematic viscosity $\nu$ is $1.14 \times 10^{-6}$, $1 \times 10^{-6}$ and $0.96 \times 10^{-6}$ m$^2$ s$^{-1}$ for 15, 20 and 15°C, respectively, and $R_0$ is the hydraulic radius resulting from the flow depth of $H = 13.5$ cm). Fish acclimatization was conducted at a discharge of 1.76 l s$^{-1}$ ($U_0 = 4.33$ cm s$^{-1}$).

| step | $T_s$–$T_e$ (min) | $Q$ (l s$^{-1}$) | $U_0$ (cm s$^{-1}$) | $Re$ (—) |
|---|---|---|---|---|
| 1 | 0–10 | 3.76 | 9.3 | 6600 |
| 2 | 10–20 | 5.77 | 14.2 | 10 110 |
| 3 | 20–30 | 7.77 | 19.2 | 13 630 |
| 4 | 30–40 | 9.77 | 24.1 | 17 150 |
| 5 | 40–50 | 11.78 | 29.1 | 20 670 |
| 6 | 50–60 | 13.78 | 34.0 | 24 180 |
| 7 | 60–70 | 15.79 | 39.0 | 27 700 |
| 8 | 70–80 | 17.79 | 43.9 | 31 220 |
| 9 | 80–90 | 19.80 | 48.9 | 34 730 |
| 10 | 90–100 | 21.80 | 53.8 | 38 250 |
| 11 | 100–110 | 23.80 | 58.8 | 41 760 |
| 12 | 110–120 | 25.80 | 63.7 | 45 270 |
| 13 | 120–130 | 27.80 | 68.7 | 48 780 |

was measured, and this included sampling points at 10–60 mm from the bed. The remaining portion of flow depth could not be measured. Filtering of the ADV data was performed using the Velocity Signal Analyser (MAJVSA v. V1.5.62) based on thresholds of signal-to-noise ratio and correlation of 20 and 70%, respectively. Despiking used the modified phase-space thresholding method by Goring & Nikora [38], revised by Wahl [39]; followed by a 12-point average spike replacement [40]. Spikes that remained after this process were identified based on standard deviation from the mean profiles [41] and excluded from the dataset. Post-processed hydrodynamic properties MAJVSA included the turbulent kinetic energy (TKE $= 0.5(\overline{u'^2} + \overline{v'^2} + \overline{w'^2})$), longitudinal and spanwise turbulence intensity

($\mathrm{TI}_u = \bar{u}'/U_0$) and ($\mathrm{TI}_v = \bar{v}'/U_0$), respectively, and Reynolds shear stresses ($\tau_{uv} = -\rho(\overline{u'v'})$), and ($\tau_{uw} = -\rho(\overline{u'w'})$) for the horizontal and vertical components, respectively, where $\rho$ is the water density (1000 kg m$^{-3}$). ($u$, $v$, $w$) and ($u'$, $v'$, $w'$) are the longitudinal, lateral and vertical ($x$, $y$, $z$) velocity and corresponding turbulent fluctuation components. Note that overbar ($\bar{\cdot}$) and bracket $\langle \cdot \rangle$ denote time averaging and spatial averaging, respectively.

## 2.5. Statistical analysis

Separate general linear models (GLMs), with Gaussian family and log-link functions, were used to examine the influence of temperature and fish standard length on burst swimming speed and time to fatigue for each species, using a Gaussian family link function. GLM models were refined by minimizing the Akaike information criterion and iterating until only significant variables ($p < 0.05$) remained, using stepwise dropping of non-significant terms from the models. Tukey honest significant difference tests were used for pairwise comparisons of means for the temperature independent variable, for 95% family-wise confidence level. Sex was also included in the models for *P. parva*. The models for sustained swimming tests accounted for hydrodynamic flow properties of mean velocities in $u$, $v$ and $w$, components ($x$, $y$, $z$) and their fluctuations, as well as the turbulence intensity, turbulent kinetic energy, and vertical and horizontal Reynolds stresses. All statistical analyses were performed using the R computing program in R Studio v. 1.2.5 [42].

# 3. Results

## 3.1. Effect of temperature on burst and sustained swimming performance

Temperature had a significant effect on the burst swimming speed of *L. gibbosus* (GLM, $p < 0.001$) but did not significantly affect that of *P. parva* (GLM, $p > 0.05$) (table 3 and figure 3). *Lepomis gibbosus* burst speeds were faster by 43% at 20°C and 53% at 25°C than those tested at 15°C (GLM, $p < 0.05$). *Pseudorasbora parva* burst speeds were on average 29% faster at 20°C but 5% slower at 25°C than at 15°C (figure 3*b*). Individual burst swimming responses of *P. parva* to temperature varied greatly, as indicated by the wide range at 20 and 25°C compared to 15°C (figure 3*a*). Standard length did not significantly influence burst swimming speed of either species (GLM, $p > 0.05$).

In the sustained swimming test, temperature significantly effected the time to fatigue ($T_f$) of both *L. gibbosus* and *P. parva* (GLM, $p < 0.05$, table 3). *Lepomis gibbosus* swam for 71% longer at 20°C and 57% longer at 25°C compared to 15°C (GLM, $p < 0.05$); although the difference between 20 and 25°C was not significant (GLM, $p = 0.089$). Similarly, *P. parva* swam for 162% longer at 20°C compared to 15°C (GLM, $p < 0.05$); but the difference in $T_f$ between 15 and 25°C was not significant (GLM, $p = 0.816$) (figure 3*b* and table 3). For both species, the effect of standard length on sustained swimming was not significant (GLM, $p = 0.5$).

## 3.2. Flume flow hydrodynamic characteristics

ADV measurements of the open channel test section were made at three locations in the centreline and three locations near the wall in vertical profiles of six points (figure 2) at all the discharges in table 2. Vertical profiles of time-averaged velocity mean $\bar{u}$, $\bar{v}$, $\bar{w}$, and fluctuations $\bar{u}'$, $\bar{v}'$, $\bar{w}'$ showed that the velocity means were similar in range and distributions along the centreline and wall (electronic supplementary material). The velocity fluctuations showed increased variance near the wall compared to the centreline. As would be expected, the streamwise components $\bar{u}$ and $\bar{u}'$ increased with increasing Reynolds number (*Re*); however, the lateral and vertical components $\bar{v}$, $\bar{w}$, $\bar{v}'$ and $\bar{w}'$ were overall higher for lower *Re*. There was a predominance of downward (negative) $\bar{w}$ velocities along the profiles for each *Re*, and as would be expected among the velocity fluctuations, $\bar{w}'$ was the lowest (electronic supplementary material).

Spatial averaging of velocities and turbulence metrics shown in figure 4 for $\langle \bar{u} \rangle$, $\langle \bar{u}' \rangle$, $\langle \mathrm{TKE} \rangle$, $\langle \tau_{uv} \rangle$ and $\langle \mathrm{TI}_v \rangle$ were performed according to the flow volume zones (shown in figure 1) for centreline locations of A, B and C and near-wall locations of D, E and F. The mean velocities $\bar{u}$ and $\bar{v}$ increased with increasing $U_0$ and were higher in the 3–6 cm elevation than the 0–3 cm near the bed, with increased scatter near the wall compared to the flume centreline. Turbulent shear stress $\tau_{uv}$ overall increased with increasing $U_0$ and was predominantly positive in the centreline, while the reverse was observed near the flume walls.

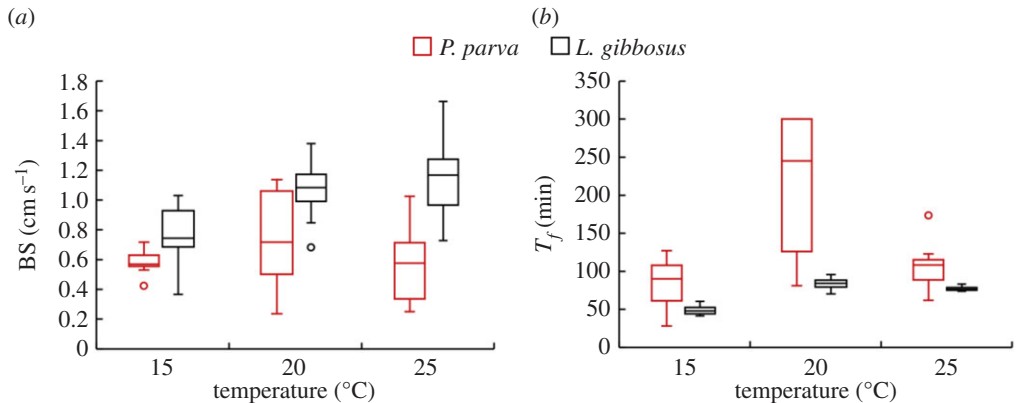

**Figure 3.** (a) Burst swimming speed BS (cm s$^{-1}$) and (b) time to fatigue $T_f$ (min) of *L. gibbosus* (filled boxes) and *P. parva* (open boxes) at 15, 20 and 25°C. Boxplots indicate, from bottom to top, minimum (lower whisker), first quartile, median, third quartile, maximum (upper whisker) and outliers (circle).

**Table 3.** Effect of temperature and other variables on burst and sustained swimming of *L. gibbosus* and *P. parva*. Independent and dependent variables are given for GLMs (degrees of freedom = 73, 70 and 66, 63 for null and residual deviance for burst and sustained swimming, respectively) and significant results are highlighted in bold.

| fish species | dependent variable | independent variables | *F*-statistic | *p*-value | post hoc comparisons/slope | *p*-value |
|---|---|---|---|---|---|---|
| *Lepomis gibbosus* | burst swimming | temperature | 15.093 | **<0.001** | 15–20 | **<0.001** |
| | | | | | 15–25 | **<0.001** |
| | | | | | 20–25 | 0.491 |
| | | standard length | 1.695 | 0.200 | | |
| | sustained swimming | temperature | 92.447 | **<0.001** | 15–20 | **<0.001** |
| | | | | | 15–25 | **<0.001** |
| | | | | | 20–25 | 0.089 |
| | | standard length | 2.556 | 0.121 | | |
| *Pseudorasbora parva* | burst swimming | temperature | 0.124 | 0.884 | | |
| | | sex | 1.483 | 0.248 | | |
| | | standard length | 0.046 | 0.833 | | |
| | sustained swimming | temperature | 6.537 | 0.004 | 15–20 | 0.027 |
| | | | | | 15–25 | 0.816 |
| | | | | | 20–25 | 0.001 |
| | | sex | 0.159 | 0.854 | | |
| | | standard length | 0.775 | 0.386 | | |

$\tau_{uw}$, on the other hand, was nearly constant for all $U_0$ and slightly higher at the vertical elevation of 3–6 than 0–3 cm. Turbulence intensity $TI_v$ was highest for the lower velocities, remained less than or equal to 2 from $U_0 = 13.18$ cm s$^{-1}$ and followed a similar distribution near the wall and in the flume centreline. The range and distribution of TKE in the centreline and near the wall were similar, which was also observed for TI, with a higher variance of these metrics for $Z < 3$ cm. The spanwise turbulence intensity $TI_v$ was consistently two to three times higher than the longitudinal turbulence intensity $TI_u$.

## 3.3. Fish swimming zone occupancy and time to fatigue

*Lepomis gibbosus* and *P. parva* sustained swimming tests were conducted using 13 velocity steps in an open channel flume where cross-sectional area velocities ranged from 9.28 to 68.65 cm s$^{-1}$,

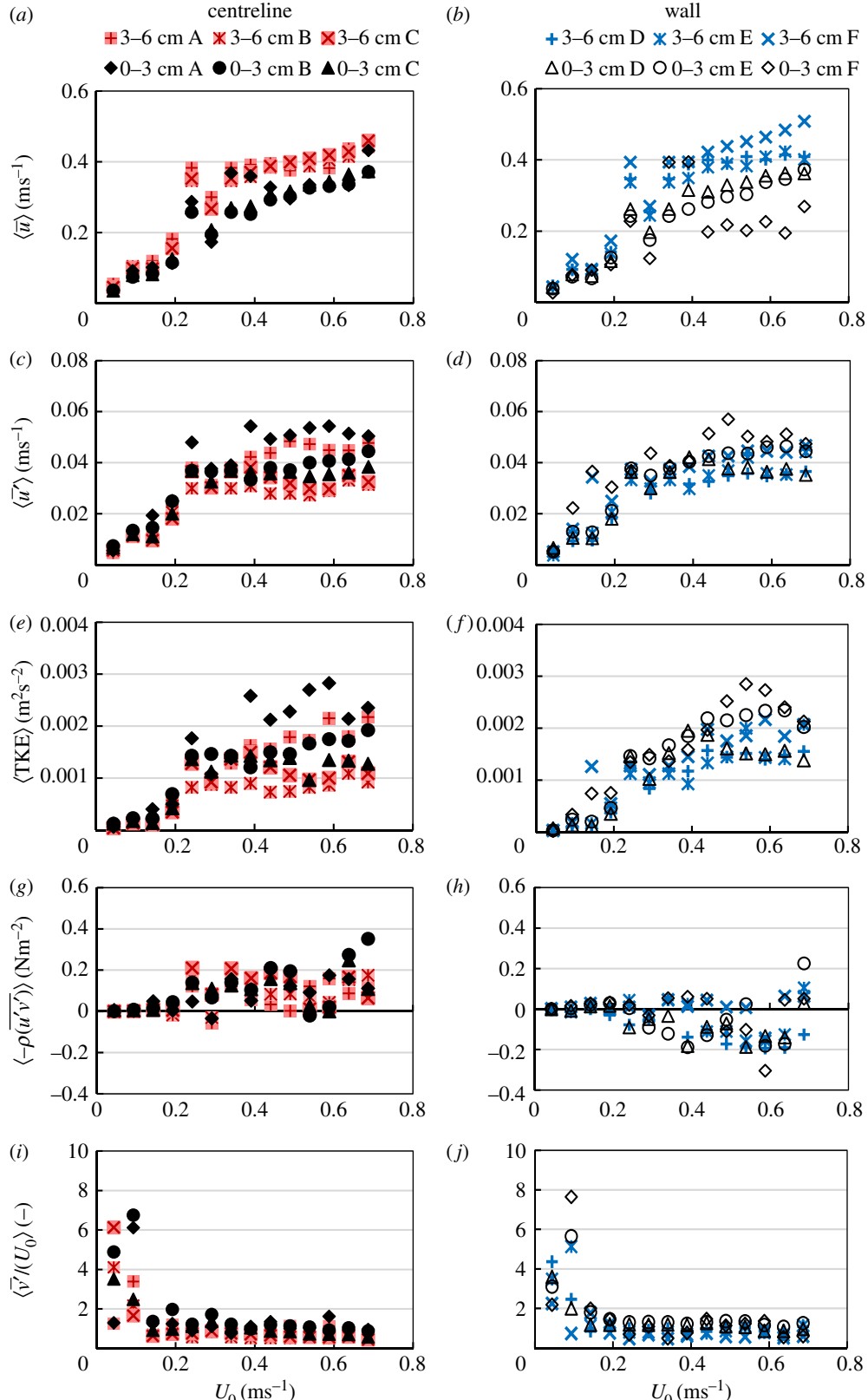

**Figure 4.** Flow velocities spatially averaged according to vertical zones of the water column at elevations of $Z = 0$–6 and 3–6 cm for all points along the centreline (A, B, C) and near the walls (D, E, F) for all cross-sectional average velocities ($U_0$) measured (shown in table 1). Displayed are longitudinal mean velocity $\langle \bar{u} \rangle$, longitudinal mean velocity fluctuation $\langle \bar{u'} \rangle$, turbulent kinetic energy $\langle TKE \rangle$, horizontal component of the Reynolds shear stress $\langle \boldsymbol{\tau}_{uv} \rangle = \langle -\rho(\overline{u'v'}) \rangle$ and spanwise turbulence intensity $\langle TI_v \rangle = \langle \bar{v'}/U_0 \rangle$ where the overline ($^{-}$) indicates time averaging and brackets $\langle \cdot \rangle$ indicate spatial averaging.

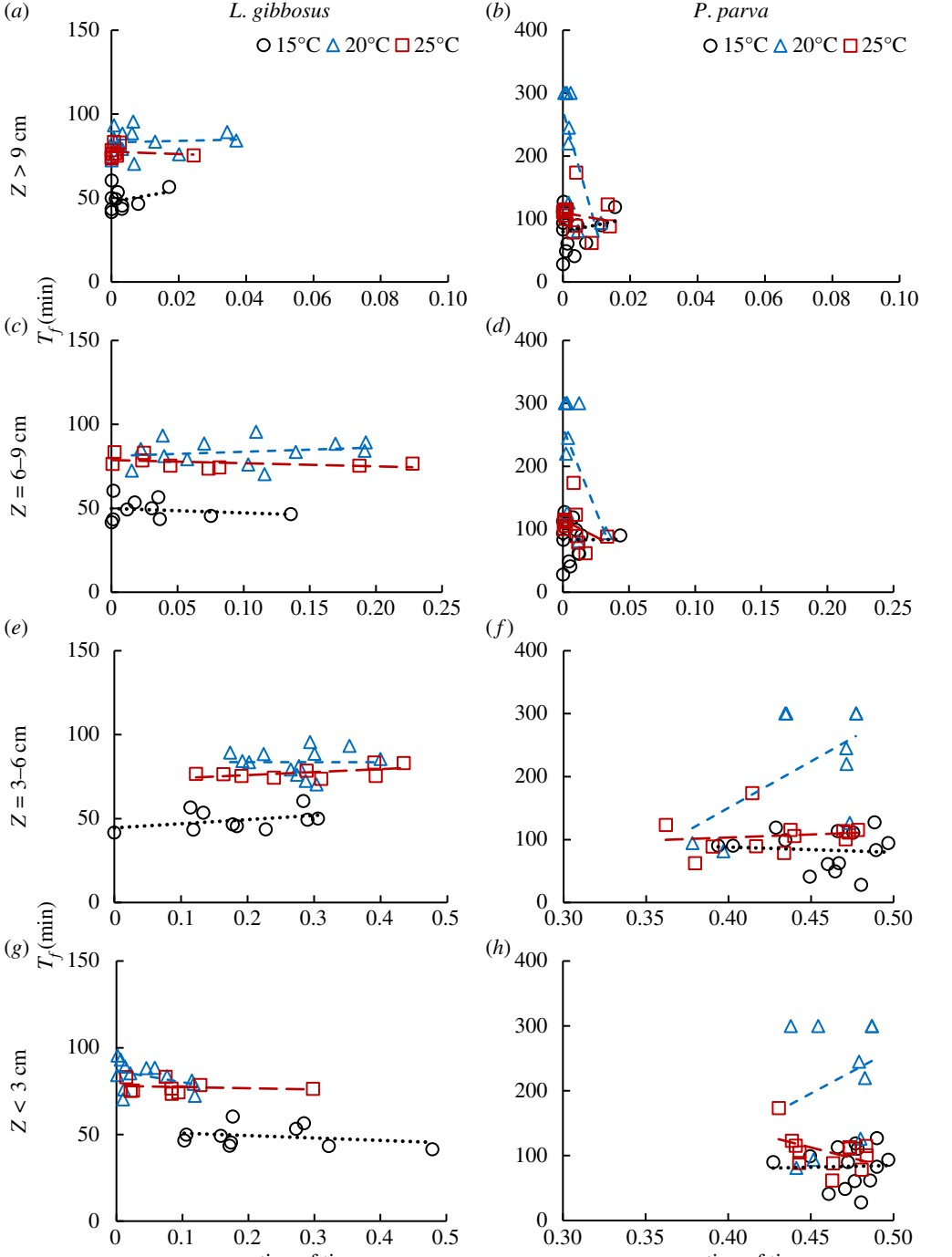

**Figure 5.** Time to fatigue ($T_f$) in relation to *L. gibbosus* and *P. parva* occupancy of the water column, in terms of the proportion of time fish spent swimming in the water column at elevations of $Z < 3$ cm, $Z = 3–6$ cm, $Z = 6–9$ cm and $Z > 9$ cm for flow temperatures of 15, 20 and 25°C.

corresponding to Reynolds numbers ranging from 6600 to 48 780 (table 2) at three water temperatures (15, 20 and 25°C). Time to fatigue changed with the amount of time fish spent in the bottom 3 cm of the flume depth for 15 and 25°C, but this was not observed at 20°C (figure 5). Time to fatigue increased with increasing time in $Z = 3–6$ cm for both species, and 6–9 cm for *L. gibbosus*, indicating a positive trend between time to fatigue and the proportion of time spent at these elevations. Figure 5 shows similar temperature-dependent variations and trends of the time to fatigue relative to the proportion of time fish spent in each zone, which were significant for *L. gibbosus* (GLM, $p < 0.05$) but not for *P. parva* (GLM, $p = 0.8$).

The distribution of fish occupancy in the water column relative to flow, temperature and cross-sectionally averaged velocity at each step is shown in figure 6, where the proportion of time fish

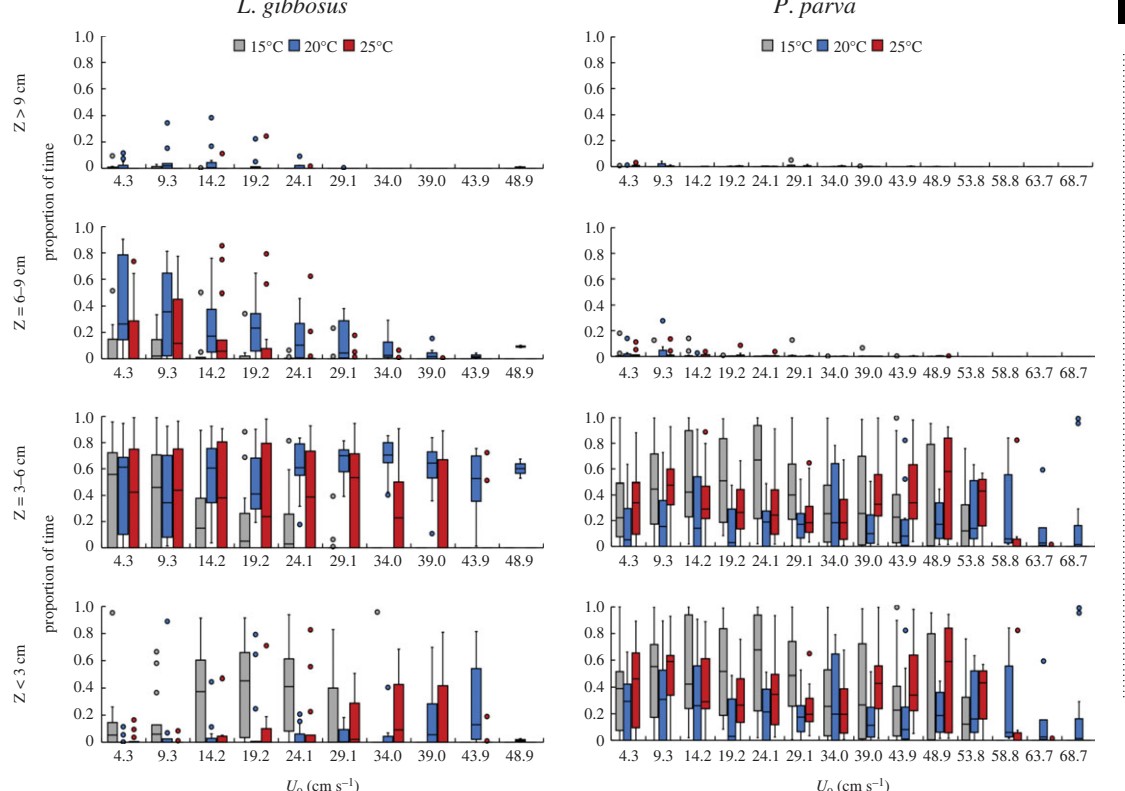

**Figure 6.** Proportion of time *L. gibbosus* and *P. parva* spent swimming in each subsection of the water column at the various flow temperatures throughout the velocity step test with $U_0$ ranging from 4.3 to 68.7 cm s$^{-1}$. Boxplots indicate, from bottom to top, minimum (lower whisker), first quartile, median, third quartile, maximum (upper whisker) and outliers (circle).

occupied the volume zones throughout the step velocity test varied in the water column with each temperature for both species. For *L. gibbosus*, the proportion of time significantly varied only between 15°C and the other two temperatures 20 and 25°C (GLM, $p < 0.001$) and their occupancy of the near-bed zones increased with increasing flow velocity. At all temperatures, *L. gibbosus* spent the least amount of time in the water column 9 cm above the flume bed (figure 6). *Pseudorasbora parva* behaved similar to *L. gibbosus*, except for minimally using the 6–9 cm layer, mainly preferring the bottom 6 cm, and not altering their occupancy considerably with increasing velocity. Both species spent the least amount of time in the water column layers above 9 cm from the bed at all three temperatures. On average, *L. gibbosus* spent less than 1% of their time near the walls, while this was up to 28% for *P. parva*.

Fish response to local mean longitudinal velocities ($\bar{u}$) and fluctuations ($\bar{u}'$) was influenced by temperature. For *L. gibbosus*, time to fatigue was negatively correlated with $\bar{u}$ and $\bar{u}'$ (GLM, $p < 0.001$), but was positively correlated with increasing lateral velocity component $\bar{v}'$ (GLM, $p < 0.05$), which suggests that *L. gibbosus* responded differently to each turbulence component. Similar relationships were observed for *P. parva*, with both components of turbulence intensity TI and Reynolds shear stresses $\tau$, as well as turbulent kinetic energy TKE found to significantly affect time to fatigue (GLM, $p < 0.05$).

Figure 7 shows the distribution of zone occupancy for both *L. gibbosus* and *P. parva* relative to longitudinal velocity and turbulent components $\bar{u}$, $\bar{u}'$ and temperature, where there are distinct differences in behaviour for both species over the three different temperatures in the proportion of time fish swam in the water column below 3 cm or in the mid-water column of 3–6 cm from the bed. At 15°C, the flow turbulent fluctuations affected the proportion of time fish spent in each flow volume zone, which increased with increasing lateral ($\bar{v}'$) and vertical ($\bar{w}'$) fluctuation components but decreased with increasing longitudinal ($\bar{u}'$) component. Furthermore, swimming closer to the bed was more prevalent with increasing longitudinal turbulence intensity TI$_u$ and spanwise Reynolds shear stress $\tau_{uv}$. The *P. parva* showed similar behaviour to that of *L. gibbosus* of using the water column differently depending on temperature throughout the swimming test. This indicates that temperature, velocity and turbulent fluctuations had a combined effect on fish's occupancy of the water column.

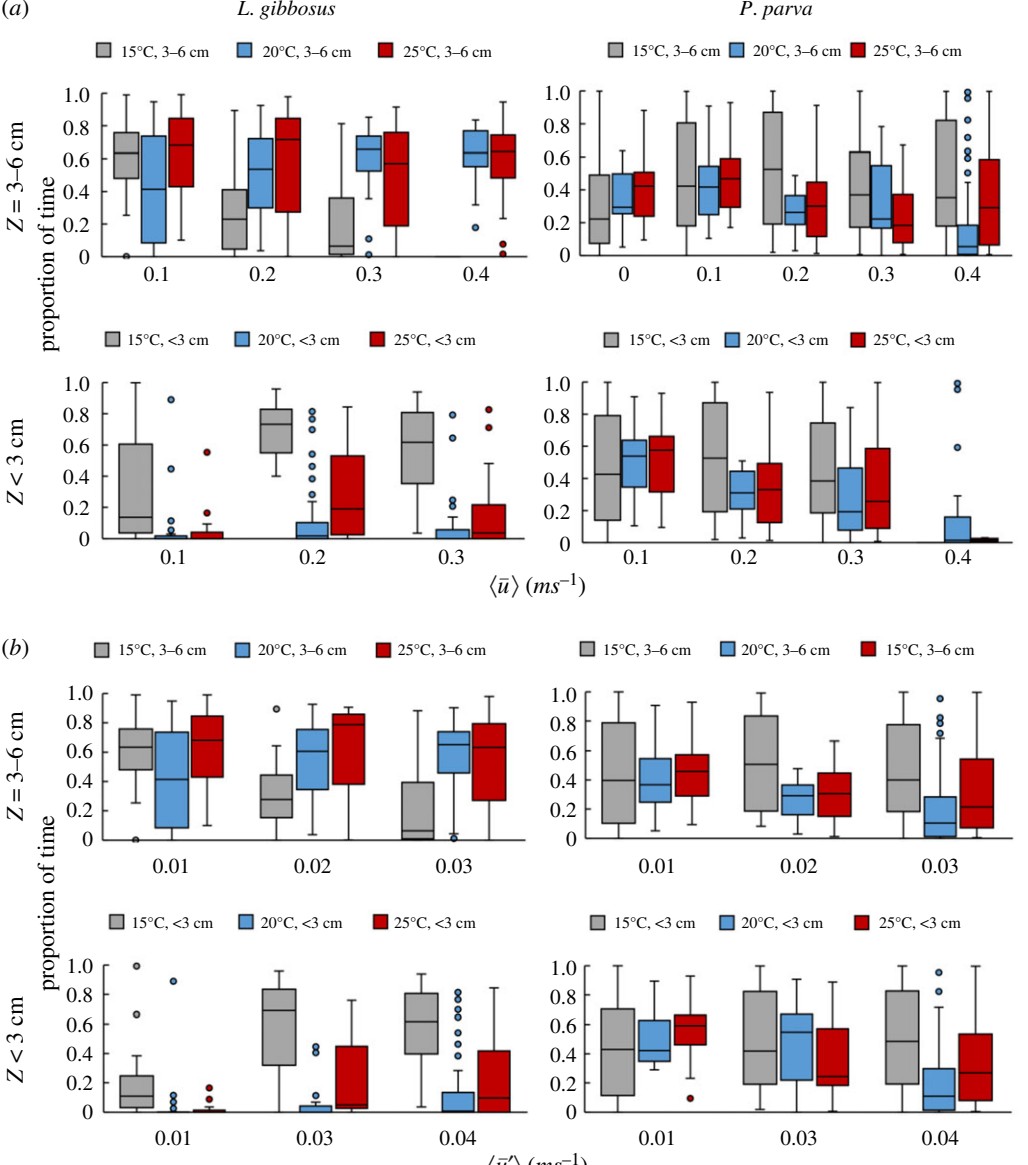

**Figure 7.** Proportion of time *L. gibbosus* and *P. parva* spent in subsections of the water column at elevations above the bed of $Z <$ 3 cm and $3 < Z > 6$ cm relative to the averaged velocities and velocity fluctuations of $\langle \bar{u} \rangle$ (a) and $\langle \bar{u}' \rangle$ (b), respectively, where the overline (¯) indicates time averaging and brackets $\langle \cdot \rangle$ indicate spatial averaging. Boxplots indicate, from bottom to top, minimum (lower whisker), first quartile, median, third quartile, maximum (upper whisker) and outliers (circle).

## 4. Discussion

Swimming performance serves as an indicator of fish endurance and ability to navigate various flow conditions, forming the basis for fish passage and other ecohydraulics measures. In addition to significant anthropogenic alterations of river habitats, seasonal variations of water thermal regimes and the predicted increases in temperature under climate change scenarios could further alter lotic ecosystems [1–5]. Considering that studies of thermal effects on fish swimming have, to date, not accounted for turbulence, and that studies evaluating turbulence–fish interactions have overlooked temperature effects, this study sought to bridge this gap, particularly with regard to the potential for range expansion of invasive non-native fishes.

Using burst and sustained swimming tests in a swimming arena and recirculating flume, respectively, we evaluated the effects of temperature, velocity and turbulence on the swimming behaviour of two invasive non-native fish species, *L. gibbosus* and *P. parva*. Temperature affected fish test zone occupancy and swimming performance, with both species exhibiting increased movement in the water column, prolonged swimming performance at warmer temperatures, and improved burst speed for *L. gibbosus*.

Temperature increases improved the burst swimming ability of *L. gibbosus*, but not that of *P. parva*. The burst swimming response of fish is believed to have evolved to allow fish to quickly move their body away from a predator [34]. Therefore, the altered response to temperature observed may have a significant impact on their ability to evade native predators. Additionally, the burst swimming response can be initiated when fish are attacking prey, and so *L. gibbosus* may benefit from a higher temperature through being able to attack faster [43].

Fish sustained swimming ability depended on temperature, which is probably related to metabolic responses, with low temperature hindering *L. gibbosus* maximum swimming speed, and at the same time reducing time to fatigue at 15°C [12,13,44,45]. This temperature response is further evidenced by the *L. gibbosus* increased occupancy of near-bed areas when the velocity steps increased under the 15°C thermal regime compared to 20 and 25°C, despite the presence of higher turbulent fluctuations at elevations less than 3 cm above the bed, while *P. parva* predominantly preferred to swim within 6 cm of the bed. Turbulent fluctuations influenced fish swimming behaviour by affecting both fish position in the water column and time to fatigue. Likewise, the elevated levels of turbulent kinetic energy and turbulence intensity were low enough not to impact on the energy saving benefits of the near-bed zone.

The predominant occupancy of the near-bed zones ($Z < 3$ cm) at 15°C might be partially explained by fish seeking to conserve energy as a physiological response to the lower temperature [12,13,19,45]. This is further attributed to the presence of relatively lower mean velocities ($\bar{u}$ and $\bar{v}$) here (figure 6), which provided a velocity refuge [16,46]. As would be expected, velocity fluctuations ($\bar{u}'$ and $\bar{v}'$) were highest near the bed (figure 6), which contributed to lowering the fish's time to fatigue at 15°C compared to 20 and 25°C. *Pseudorasbora parva*, spent more time near the flume bed with increasing step velocity compared to *L. gibbosus* (figure 7) as the energetic costs of swimming could be increased by the increase in flow velocity [12,46,47], which led the fish to seek relatively lower mean velocities. The varied temperature response observed between both species is probably caused by their different maturity stages, physiological optima reflecting their native temperature ranges.

A distinction was made between the flume centreline and near wall due to the variations in hydrodynamic properties of the centre and sidewalls of open channels [48] and the tendency for fish to seek velocity refuges near walls [16]. However, in the current study, although both species are bentho-pelagic [28,49], *P. parva* spent more time (28%) than *L. gibbosus* (less than 1%) swimming in near-wall areas, which suggest that the former used the near-wall areas as a flow refuge [14]. This might be due to the similarities in distribution of local velocity and turbulence metrics between the flume centreline and the near-wall areas. Importantly, differences in body shape between the two species could also affect occupancy of the test area zones; the *P. parva* being slimmer and sleeker than the more laterally compressed, deeper-bodied *L. gibbosus*. The flow hydrodynamic properties indicate the presence of secondary currents in the channel (figure 4) (electronic supplementary material), which, in conjunction with their body shape, might have deterred *L. gibbosus* from occupying the near-wall areas. The secondary currents are evident in the horizontal Reynolds shear stress as its direction near the flume wall is opposite that of the centreline (figure 4), indicating flow circulation in the *YZ* plane, characteristic of secondary flows in open channels [48].

In summary, under three temperature regimes, the occupancy of test zone areas of *L. gibbosus* and *P. parva* was highly dependent upon temperature, with swimming performance increasing at elevated temperatures. Evaluation of thermal regimes when studying fish-flow dynamics is recommended, since fish swimming behaviour will be impacted, and this is likely to vary depending on fish species. Thus, this study has demonstrated that temperature is an important factor affecting the swimming performance of invasive non-native fish species. These findings are of particular importance when considering the potential effects of predicted climate change scenarios and how these may alter the dispersal opportunities and subsequent establishment success of invasive non-native fishes in the wild.

Data accessibility. Data for this manuscript are available in electronic supplementary material.

Authors' contributions. J.C., C.A.M.E.W., J.R.T. and V.M.: conceptualization and experimental design; J.R.T., V.M. and J.N.: data collection analysis and visualization; F.C.W. and D.G.D.: resources; V.M., J.R.T., J.C. and C.A.M.E.W.: writing draft. All authors contributed to reviewing and editing the manuscript.

Competing interests. We declare we have no competing interests.

Funding. This work was supported by a Cardiff University International PhD studentship to V.M.; Coleg Cymraeg Cenedlaethol funding for J.R.T.; and Cardiff University Research Opportunity Project 2018 (CUROP) and Placement Training Year at Cardiff University School of Engineering 2017/2018 (PTY) for J.N.

Acknowledgements. We thank Elin Dumont for technical assistance, and the three anonymous reviewers for their invaluable comments.

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
