## [Peer Review File · Royal Society Open Science]

Review History

RSOS-201516.R0 (Original submission)

Review form: Reviewer 1

Is the manuscript scientifically sound in its present form?

Yes

Are the interpretations and conclusions justified by the results?

Yes

Is the language acceptable?

Yes

Do you have any ethical concerns with this paper?

No

Have you any concerns about statistical analyses in this paper?

No

Recommendation?

Accept with minor revision (please list in comments)

Comments to the Author(s)

In the present study, the authors investigated the effects of temperature, velocity and turbulence on swimming performance of two fishes, invasive to the UK (pumpkinseed and topmouth gudgeon). They tested swimming performance in a flume at three different set temperature while varying velocity within each setting (which also led to changes in turbulence in different parts of the experimental arena). They found that compared to the effects on temperature, effects of velocity and turbulence were relatively small. Specifically, swimming performance largely increased with increasing temperatures. They then discuss these results also with respect to how the invasiveness of these fish might be exacerbated by increasing temperatures resulting from global climate change.

I found this manuscript well written and the methodology appropriate. I did not find any major issues with the manuscript, study design, analysis or interpretation but discovered a few smaller issues that will need to be corrected as part of a revision.

Minor comments:

- 1 - A streamlining of the usage of non-native and invasive throughout would be beneficial. Currently, this seems to jump around a bit. For example, in lines 68-69, the pumpkin seed is labelled as non-native while the topmouth gudgeon is labelled as invasive? Aren't they both invasive to the UK?
- 2 - Methods, lines 93-94: Regarding the measurements of standard length and weight - What was the measurement level? Was SL quantified to the nearest mm, or even to some nearest decimal of mm? What about weight? Similarly, regarding the mention of a photograph having been taken, please specify what kind of camera this was.
- 3 - Methods, lines 110-112: This needs a bit more information as the numbers in this sentence do not add up. If the inner arena was only 10cm high, then the water depth could not have been 12cm. I am assuming, from looking at Figure 1, that the outer container was much higher than the inner container, so that the water level surpassed the height of the inner arena? This needs to be clarified as I am only guessing this.
- 4 - Results, lines 179-182: This wording here is a bit confusing, as the posthoc tests listed in Table 3 do not always show that the differences that are mentioned here in this part of the text were actually significant. It would be good to also indicate somewhere here in the text which pairwise temp comparisons were actually significant and which were not.
- 5 - Figure 3: This could just be an artifact of the file upload, but currently this figure looks like it is relatively low resolution compared to the other figures. If true, please increase resolution. Also, the font format changes between both panels (also in some of the later figures 4, 5 and 6). Please adjust to the same format in both/all panels.

Specific comments:

Abstract, line 24: Please switch 'on as' to 'as on'

Introduction, lines 35-36: Some conjunctive adverb or similar would be good here, given that this sentence appears to be meant as a single-sentence statement rather than the lead in to an in-depth paragraph of climate change. I suggest something along the lines of: Climate change, on the other hand, impacts all environments, significantly...

Introduction, line 69: Here and throughout - please do not abbreviate the genus name at the beginning of a sentence.

Introduction, line 72: Please change to 'is subject to'.

Methods, line 163: Why was 'sex' only included in the models for *P. parva*? I assume this was because *L. gibbosus* lacks sexual dimorphism? Please explain and justify.

Results, line 173: I am not a native speaker, but this reads odd to me. Maybe rephrase to: '...were on average 29% faster at 20°C but 5% slower at 25°C, than at 15°C.'

Discussion, lines 255-256: There are a lot of 'ands' in this sentence which makes it awkward to read; please rephrase.

References, line 341: Please remove '(80-).' from the entry.

References, lines 345-347: Please provide journal name, volume and page range.

References, line 378: Please put '*Galaxias maculatus*' in italics.

References, lines 441-442: Please provide page range or e-article number, and please put volume number in bold.

References, line 458: Please delete 'LP' from the middle of the page range.

References, lines 470-471: Please correct the title of this publication, something went wrong there. Also, the page range provided is not correct.

Table 3, lines 489-491: What were the degrees of freedom for these models?

Figure 1 legend, line 495: Please change 'no to scale' to 'not to scale'.

Review form: Reviewer 2

Is the manuscript scientifically sound in its present form?

Yes

Are the interpretations and conclusions justified by the results?

No

Is the language acceptable?

Yes

Do you have any ethical concerns with this paper?

No

Have you any concerns about statistical analyses in this paper?

No

Recommendation?

Accept with minor revision (please list in comments)

Comments to the Author(s)

This is a nicely designed study investigating the effects of water temperature, flow velocity and turbulence on swimming performance in two invasive fish species. The strength of this study is the detail of the design and its successful implementation to capture some intricate measurements. The authors were able to quantify swimming performance and behaviour (area of water column occupied) and showed an interplay with thermal conditions and hydrodynamic flow characteristics. I think this work is a valuable addition to the literature and clearly demonstrates better modelling of real-world conditions provides better understanding. The whole manuscript is written in a comprehensive and engaging manner and Table 3 is an excellent way of presenting the statistical results. I have a few comments:

1. Please listen to your statistics – temperature had no effect on burst speed swimming in topmouth gudgeon, *Pseudorasbora parva*, yet in the results, discussion and abstract the authors continually state that burst speed swimming was positively affected by temperature for both species. This is not true and the text should be rewritten to make this clear. You have already correctly referred to these results in lines 170-171, but please remove the text “ *P. parva* burst speeds were on average 29% faster but 5% slower at 20 and 25°C, respectively, than at 15°C (Fig. 3B)” note- this is actually Fig. 3A. Please also amend the rest of the manuscript to better reflect your results.
2. Continuing the theme that *Lepomis gibbosus* and *P. parva* showed different results, I would like to suggest the different ecologies of the two species should be considered in your interpretation of the results. *P. parvus* is a temperate species (FishBase provides a temperature range of 5 - 22°C) whilst *L. gibbosus* is subtropical (4 - 30°C). The varying effects of different temperatures on burst speed in *P. parvus* may be explained by the upper temperature tested (25°C) exceeding a pejus temperature for this species.
3. The figures are not particularly intuitive and could be simplified to convey the key information. Figure 2 takes a few reads to fully understand it – what does the triangle with three lines underneath it represent? The data are complex, so difficult to visualise but I do feel they could be displayed better in places, especially in Figure 4. In addition, the legend for Figure 5 is incomplete (I am guessing the red squares are 25°C data) and these graphs might work better as regressions plus confidence intervals, rather than showing all the data. Please also continue your convention of using the species name (*L. Gibbosus* and *P. parva*), not the common name, in the figure legend.
4. There is some inconsistency in use of symbols: l or L for litre (e.g. lines 124 and 128).

Review form: Reviewer 3**Is the manuscript scientifically sound in its present form?**

No

Are the interpretations and conclusions justified by the results?

No

Is the language acceptable?

Yes

Do you have any ethical concerns with this paper?

No

Have you any concerns about statistical analyses in this paper?

Yes

Recommendation?

Major revision is needed (please make suggestions in comments)

Comments to the Author(s)

I have reviewed the manuscript RSOS-201516 Temperature surpasses the effects of velocity and turbulence on swimming performance of two invasive non-native fish species by Muhawenimana et al. The manuscript characterises the swimming performances of two freshwater fish species in response to environmental temperature. The authors posit that the performance characteristics of the two species, which are non-native in the UK, could facilitate range expansions with climate change. The manuscript is well written, however there are several problems with the experimental design and reporting of methodology that prevent me from recommending it for publication at this time.

1. There are no ecological data provided on the environments currently inhabited by the two species in the study/sampling location. Thermal data are particularly important in this respect. Do the test temperatures selected represent realistic thermal regimes experienced in their current locations and how well do they reflect proposed thermal regimes under the various IPCC warming scenarios? The findings should be put into more of an ecological context.
2. There is very little coverage of the thermal physiology of swimming in fish, and more importantly, how different species may deal with thermal variability in their environments. This experimental design only allows for the thermal sensitivity of the performance trait over the thermal range to be examined. There is no consideration of the potential for thermal plasticity (acclimation/acclimatization), adaptation or behavioural changes to offset, or possibly, amplify the effects of climate warming on fish performance.
3. The experiment attempts to link behavioural attributes ('preferences') with turbulence and flow velocity and swimming performance. Firstly, all these experiments were conducted in a long glass swim channel (I think, however details about flume are lacking) with flow straighteners all of which are designed to minimise turbulence in the fluid flow stream. Although I understand that there is some turbulence in the swim channel particularly at the channel walls where the fluid interacts with the structure and this increases with fluid velocity, this level of turbulence would be miniscule compared to the turbulence in natural environments. Velocity and turbulence are highly correlated in this experiment as well. More importantly, the authors have not set out to empirically test how hydrodynamics affects swimming performance and instead are relying on correlations between fish presence and turbulence/velocity to inform their position. I'm not sure that this is a valid or particularly useful way to test this idea.
4. There are no hypotheses presented. I do not believe that the gap in knowledge that the MS is attempting to address has been clearly articulated.
5. Statistics and methods need greater attention. There are insufficient details provided about the flume and its operation, there are no details provided on the generation of the thermal treatments for the sustained swimming tests, there are no details about how behavioural data were collected from videos and how these may have been analysed statistically. How long were fish given to adjust to thermal conditions before testing? Were they fasted before testing? What

time of day was testing conducted? Were fish re-swum for any tests or were unique individuals used for each metric, if so were repeated measures tests considered? Turbulence and velocity metrics are likely to all be highly correlated, so the statistical treatment of these variables needs further consideration (multicollinearity). I would like to see the statistical models more fully detailed, particularly how the hydrodynamic variables were considered as they related to performance/behaviour.

6. There is a disjuncture between the premise for the manuscript established in the introduction and the conclusions drawn in the discussion. The introduction sets the MS up to be primarily about invasive species management and projected range expansions with climate change, while the discussion is dominated largely by consideration of swimming position in the flume and effects of body shape and temperature thereon. While there is a place for the authors to consider the implications of their research for invasive species management, the design of the study does not really support this being the dominant focus of the manuscript, and so the introduction needs to be suitably revised. The study is primarily about the thermal sensitivity of performance (and to a lesser extent, behaviour) in these two fish species. If the authors want to take the invasive species angle, then they should be to be comparing performance and behaviour across native and other non-native species (in a phylogenetically considered manner).

Minor points:

1. line 39: reference required after 'drivers'.
2. Line 62-67: The authors need to broaden their literature search criteria or reduce the generality of these statements. There are countless flume studies looking at the effects of temperature on fish swimming performance – the majority of these manipulate fluid flow velocity to measure swimming performance.
3. Ln 68: insert 'freshwater fish' before 'species'
4. Ln 113: how did you deal with the three measures? Averaged? Fastest?
5. Ln 116: filming speed (frame grab rate)? Describe how fish movement was actually measured. Did you only use recordings where you got an actual 'C' start? Report values in SI units (m s⁻¹).
6. Section 2.3 – please provide a more detailed description of flume design (construction material, method for generation of fluid movement, baffling, header tanks, downstream gates etc.). Description of generation of thermal treatments is lacking in this section. How were fish introduced into the flume? How did you define fatigue? Please describe how behaviour metrics were analysed (define each of the behaviour metrics).
7. Line 145: grammar
8. line 156: what is ρ ? Were measurements repeated at each velocity increment and at each temperature.
9. Ln 163 – 166: temperature? Please define all model response variables and independent variables. Post-hoc tests? What packages/functions were used? Citations for these?
10. Section 3.3. There is so much behavioural methodology missing that don't feel that I can appropriately assess this section.
11. Ln 263-268 – these are a repetition of the results
12. Ln 285: what is meant by size effects?
13. Table 2 – you don't need the t_e and t_s columns. It's enough to say that the time increment at each velocity was 10 min.
14. Ln 485 what was the value for water kinematic velocity?
15. Line 506 & 520: Please provide the definitions for the abbreviations in these figure legends. Legends should be able to stand on their own without reference to the main body text.

Decision letter (RSOS-201516.R0)

Dear Ms Muhawenimana

The Editors assigned to your paper RSOS-201516 "Temperature surpasses the effects of velocity and turbulence on swimming performance of two invasive non-native fish species" have now received comments from reviewers and would like you to revise the paper in accordance with the reviewer comments and any comments from the Editors. Please note this decision does not guarantee eventual acceptance.

Please submit your revised manuscript and required files (see below) no later than 21 days from today's (ie 16-Nov-2020) date. Note: the ScholarOne system will 'lock' if submission of the revision is attempted 21 or more days after the deadline. If you do not think you will be able to meet this deadline please contact the editorial office immediately.

on behalf of Prof Pete Smith (Subject Editor)
openscience@royalsociety.org

Associate Editor Comments to Author:

Please carefully revise your paper in response to feedback you have received - a revision will be sent to at least one of the reviewers for further oversight, and you should be aware that the journal does not routinely permit multiple rounds of revision.

Reviewer comments to Author:

Reviewer: 1

Comments to the Author(s)

In the present study, the authors investigated the effects of temperature, velocity and turbulence on swimming performance of two fishes, invasive to the UK (pumpkinseed and topmouth gudgeon). They tested swimming performance in a flume at three different set temperature while varying velocity within each setting (which also led to changes in turbulence in different parts of the experimental arena). They found that compared to the effects on temperature, effects of velocity and turbulence were relatively small. Specifically, swimming performance largely increased with increasing temperatures. They then discuss these results also with respect to how the invasiveness of these fish might be exacerbated by increasing temperatures resulting from global climate change.

I found this manuscript well written and the methodology appropriate. I did not find any major issues with the manuscript, study design, analysis or interpretation but discovered a few smaller issues that will need to be corrected as part of a revision.

Minor comments:

- 1 - A streamlining of the usage of non-native and invasive throughout would be beneficial. Currently, this seems to jump around a bit. For example, in lines 68-69, the pumpkin seed is labelled as non-native while the topmouth gudgeon is labelled as invasive? Aren't they both invasive to the UK?
- 2 - Methods, lines 93-94: Regarding the measurements of standard length and weight - What was the measurement level? Was SL quantified to the nearest mm, or even to some nearest decimal of mm? What about weight? Similarly, regarding the mention of a photograph having been taken, please specify what kind of camera this was.
- 3 - Methods, lines 110-112: This needs a bit more information as the numbers in this sentence do not add up. If the inner arena was only 10cm high, then the water depth could not have been 12cm. I am assuming, from looking at Figure 1, that the outer container was much higher than the inner container, so that the water level surpassed the height of the inner arena? This needs to be clarified as I am only guessing this.
- 4 - Results, lines 179-182: This wording here is a bit confusing, as the posthoc tests listed in Table 3 do not always show that the differences that are mentioned here in this part of the text were actually significant. It would be good to also indicate somewhere here in the text which pairwise temp comparisons were actually significant and which were not.
- 5 - Figure 3: This could just be an artifact of the file upload, but currently this figure looks like it is relatively low resolution compared to the other figures. If true, please increase resolution. Also, the font format changes between both panels (also in some of the later figures 4, 5 and 6). Please adjust to the same format in both/all panels.

Specific comments:

Abstract, line 24: Please switch 'on as' to 'as on'

Introduction, lines 35-36: Some conjunctive adverb or similar would be good here, given that this sentence appears to be meant as a single-sentence statement rather than the lead in to an in-depth

paragraph of climate change. I suggest something along the lines of: Climate change, on the other hand, impacts all environments, significantly...

Introduction, line 69: Here and throughout - please do not abbreviate the genus name at the beginning of a sentence.

Introduction, line 72: Please change to 'is subject to'.

Methods, line 163: Why was 'sex' only included in the models for *P. parva*? I assume this was because *L. gibbosus* lacks sexual dimorphism? Please explain and justify.

Results, line 173: I am not a native speaker, but this reads odd to me. Maybe rephrase to: '...were on average 29% faster at 20°C but 5% slower at 25°C, than at 15°C.'

Discussion, lines 255-256: There are a lot of 'ands' in this sentence which makes it awkward to read; please rephrase.

References, line 341: Please remove '(80-).' from the entry.

References, lines 345-347: Please provide journal name, volume and page range.

References, line 378: Please put '*Galaxias maculatus*' in italics.

References, lines 441-442: Please provide page range or e-article number, and please put volume number in bold.

References, line 458: Please delete 'LP' from the middle of the page range.

References, lines 470-471: Please correct the title of this publication, something went wrong there. Also, the page range provided is not correct.

Table 3, lines 489-491: What were the degrees of freedom for these models?

Figure 1 legend, line 495: Please change 'no to scale' to 'not to scale'.

Reviewer: 2

Comments to the Author(s)

This is a nicely designed study investigating the effects of water temperature, flow velocity and turbulence on swimming performance in two invasive fish species. The strength of this study is the detail of the design and its successful implementation to capture some intricate measurements. The authors were able to quantify swimming performance and behaviour (area of water column occupied) and showed an interplay with thermal conditions and hydrodynamic flow characteristics. I think this work is a valuable addition to the literature and clearly demonstrates better modelling of real-world conditions provides better understanding. The whole manuscript is written in a comprehensive and engaging manner and Table 3 is an excellent way of presenting the statistical results. I have a few comments:

1. Please listen to your statistics – temperature had no effect on burst speed swimming in topmouth gudgeon, *Pseudorasbora parva*, yet in the results, discussion and abstract the authors continually state that burst speed swimming was positively affected by temperature for both species. This is not true and the text should be rewritten to make this clear. You have already correctly referred to these results in lines 170-171, but please remove the text “ *P. parva* burst

speeds were on average 29% faster but 5% slower at 20 and 25°C, respectively, than at 15°C (Fig. 3B)” note- this is actually Fig. 3A. Please also amend the rest of the manuscript to better reflect your results.

2. Continuing the theme that *Lepomis gibbosus* and *P. parva* showed different results, I would like to suggest the different ecologies of the two species should be considered in your interpretation of the results. *P. parvus* is a temperate species (FishBase provides a temperature range of 5 - 22°C) whilst *L. gibbosus* is subtropical (4 - 30°C). The varying effects of different temperatures on burst speed in *P. parvus* may be explained by the upper temperature tested (25°C) exceeding a pejus temperature for this species.

3. The figures are not particularly intuitive and could be simplified to convey the key information. Figure 2 takes a few reads to fully understand it – what does the triangle with three lines underneath it represent? The data are complex, so difficult to visualise but I do feel they could be displayed better in places, especially in Figure 4. In addition, the legend for Figure 5 is incomplete (I am guessing the red squares are 25°C data) and these graphs might work better as regressions plus confidence intervals, rather than showing all the data. Please also continue your convention of using the species name (*L. Gibbosus* and *P. parva*), not the common name, in the figure legend.

4. There is some inconsistency in use of symbols: l or L for litre (e.g. lines 124 and 128).

Reviewer: 3

Comments to the Author(s)

I have reviewed the manuscript RSOS-201516 Temperature surpasses the effects of velocity and turbulence on swimming performance of two invasive non-native fish species by Muhawenimana et al. The manuscript characterises the swimming performances of two freshwater fish species in response to environmental temperature. The authors posit that the performance characteristics of the two species, which are non-native in the UK, could facilitate range expansions with climate change. The manuscript is well written, however there are several problems with the experimental design and reporting of methodology that prevent me from recommending it for publication at this time.

1. There are no ecological data provided on the environments currently inhabited by the two species in the study/sampling location. Thermal data are particularly import in this respect. Do the test temperatures selected represent realistic thermal regimes experienced in their current locations and how well do they reflect proposed thermal regimes under the various IPCC warming scenarios? The findings should be put into more of an ecological context.

2. There is very little coverage of the thermal physiology of swimming in fish, and more importantly, how different species may deal with thermal variability in their environments. This experimental design only allows for the thermal sensitivity of the performance trait over the thermal range to be examined. There is no consideration of the potential for thermal plasticity (acclimation/acclimatization), adaptation or behavioural changes to offset, or possibly, amplify the effects of climate warming on fish performance.

3. The experiment attempts to link behavioural attributes ('preferences') with turbulence and flow velocity and swimming performance. Firstly, all these experiments were conducted in a long glass swim channel (I think, however details about flume are lacking) with flow straighteners all of which are designed to minimise turbulence in the fluid flow stream. Although I understand that there is some turbulence in the swim channel particularly at the channel walls where the fluid interacts with the structure and this increases with fluid velocity, this level of turbulence would be miniscule compared to the turbulence in natural environments. Velocity and turbulence are highly correlated in this experiment as well. More importantly, the authors have not set out to

empirically test how hydrodynamics affects swimming performance and instead are relying on correlations between fish presence and turbulence/velocity to inform their position. I'm not sure that this is a valid or particularly useful way to test this idea.

4. There are no hypotheses presented. I do not believe that the gap in knowledge that the MS is attempting to address has been clearly articulated.

5. Statistics and methods need greater attention. There are insufficient details provided about the flume and its operation, there are no details provided on the generation of the thermal treatments for the sustained swimming tests, there are no details about how behavioural data were collected from videos and how these may have been analysed statistically. How long were fish given to adjust to thermal conditions before testing? Were they fasted before testing? What time of day was testing conducted? Were fish re-swum for any tests or were unique individuals used for each metric, if so were repeated measures tests considered? Turbulence and velocity metrics are likely to all be highly correlated, so the statistical treatment of these variables needs further consideration (multicollinearity). I would like to see the statistical models more fully detailed, particularly how the hydrodynamic variables were considered as they related to performance/behaviour.

6. There is a disjuncture between the premise for the manuscript established in the introduction and the conclusions drawn in the discussion. The introduction sets the MS up to be primarily about invasive species management and projected range expansions with climate change, while the discussion is dominated largely by consideration of swimming position in the flume and effects of body shape and temperature thereon. While there is a place for the authors to consider the implications of their research for invasive species management, the design of the study does not really support this being the dominant focus of the manuscript, and so the introduction needs to be suitably revised. The study is primarily about the thermal sensitivity of performance (and to a lesser extent, behaviour) in these two fish species. If the authors want to take the invasive species angle, then they should be to be comparing performance and behaviour across native and other non-native species (in a phylogenetically considered manner).

Minor points:

1. line 39: reference required after 'drivers'.

2. Line 62-67: The authors need to broaden their literature search criteria or reduce the generality of these statements. There are countless flume studies looking at the effects of temperature on fish swimming performance – the majority of these manipulate fluid flow velocity to measure swimming performance.

3. Ln 68: insert 'freshwater fish' before 'species'

4. Ln 113: how did you deal with the three measures? Averaged? Fastest?

5. Ln 116: filming speed (frame grab rate)? Describe how fish movement was actually measured. Did you only use recordings where you got an actual 'C' start? Report values in SI units (m s⁻¹).

6. Section 2.3 – please provide a more detailed description of flume design (construction material, method for generation of fluid movement, baffling, header tanks, downstream gates etc.).

Description of generation of thermal treatments is lacking in this section. How were fish introduced into the flume? How did you define fatigue? Please describe how behaviour metrics were analysed (define each of the behaviour metrics).

7. Line 145: grammar

8. line 156: what is ρ ? Were measurements repeated at each velocity increment and at each temperature.

9. Ln 163 – 166: temperature? Please define all model response variables and independent variables. Post-hoc tests? What packages/functions were used? Citations for these?

10. Section 3.3. There is so much behavioural methodology missing that don't feel that I can appropriately assess this section.

11. Ln 263-268 – these are a repetition of the results

12. Ln 285: what is meant by size effects?

13. Table 2 – you don't need the te and ts columns. It's enough to say that the time increment at each velocity was 10 min.

14. In 485 what was the value for water kinematic velocity?
 15. Line 506 & 520: Please provide the definitions for the abbreviations in these figure legends. Legends should be able to stand on their own without reference to the main body text.

===PREPARING YOUR MANUSCRIPT===

Your revised paper should include the changes requested by the referees and Editors of your manuscript. You should provide two versions of this manuscript and both versions must be provided in an editable format:
 one version identifying all the changes that have been made (for instance, in coloured highlight, in bold text, or tracked changes);
 a 'clean' version of the new manuscript that incorporates the changes made, but does not highlight them. This version will be used for typesetting if your manuscript is accepted.
 Please ensure that any equations included in the paper are editable text and not embedded images.

===PREPARING YOUR REVISION IN SCHOLARONE===

- 1) One version identifying all the changes that have been made (for instance, in coloured highlight, in bold text, or tracked changes);
 - 2) A 'clean' version of the new manuscript that incorporates the changes made, but does not highlight them.
 - An individual file of each figure (EPS or print-quality PDF preferred [either format should be produced directly from original creation package], or original software format).
 - An editable file of each table (.doc, .docx, .xls, .xlsx, or .csv).
 - An editable file of all figure and table captions.
- Note: you may upload the figure, table, and caption files in a single Zip folder.
- Any electronic supplementary material (ESM).
 - If you are requesting a discretionary waiver for the article processing charge, the waiver form must be included at this step.
 - If you are providing image files for potential cover images, please upload these at this step, and inform the editorial office you have done so. You must hold the copyright to any image provided.
 - A copy of your point-by-point response to referees and Editors. This will expedite the preparation of your proof.

- Ensure that your data access statement meets the requirements at <https://royalsociety.org/journals/authors/author-guidelines/#data>. You should ensure that you cite the dataset in your reference list. If you have deposited data etc in the Dryad repository, please include both the 'For publication' link and 'For review' link at this stage.
- If you are requesting an article processing charge waiver, you must select the relevant waiver option (if requesting a discretionary waiver, the form should have been uploaded at Step 3 'File upload' above).
- If you have uploaded ESM files, please ensure you follow the guidance at <https://royalsociety.org/journals/authors/author-guidelines/#supplementary-material> to include a suitable title and informative caption. An example of appropriate titling and captioning may be found at https://figshare.com/articles/Table_S2_from_Is_there_a_trade-off_between_peak_performance_and_performance_breadth_across_temperatures_for_aerobic_sc_ope_in_teleost_fishes_/3843624.

Author's Response to Decision Letter for (RSOS-201516.R0)

See Appendix A.

RSOS-201516.R1 (Revision)

Review form: Reviewer 1

Is the manuscript scientifically sound in its present form?

Yes

Are the interpretations and conclusions justified by the results?

Yes

Is the language acceptable?

Yes

Do you have any ethical concerns with this paper?

No

Have you any concerns about statistical analyses in this paper?

No

Recommendation?

Accept with minor revision (please list in comments)

Comments to the Author(s)

I wanted to thank the authors for their careful revisions based on my previous set of comments. I found the paper much improved but also noticed that not all changes had always been done as stated. I therefore list below a few follow-up issues that arose during the revisions.

1 - I had originally asked for clarification on how certain measurements were done (listed as comment 1.2 in the reply to comments). While the authors replied with most of the needed details in the reply document (thank you), they only added some of this to the manuscript. For example, the manuscript still does not state that photographs were taken with an iPhone camera. This needs to be added, as well as which model that iPhone was (since several older models do not really have cameras with a very good resolution, which could impact on the quality of the pictures). Also, they now clarify in the revised manuscript that weight was measured to two decimal places (again, thank you) but still fail to mention what unit of measure was measured to two decimal places. Was this g? Please add this information to the manuscript.

2 - I had mentioned that the font size and format changed between panels within figures. While this was corrected for some (thank you), it was not corrected for all figures. Please also adjust this for all panels in figure 5.

3 - Thank you for clarifying why you only included 'sex' as a factor for one species but not the other. However, does this mean that *Lepomis* were juveniles but *Pseudoraspora* were adults? If so, then some of the differences you found between species could have easily been also the result of testing juveniles versus adults. This needs to be directly addressed in the discussion.

4 - Thank you for adding the degrees of freedom to the Table 3 header. However, you now list a single degree of freedom for each type of model, is this the error degree of freedom? For different terms in a Gaussian GLM you have to state two different degrees of freedom, the DF associated with that term and the overall error DF for the model (Gotelli & Ellison, 2004, *A Primer of Ecological Statistics*, Sinauer Associates; or see here: <http://psych.colorado.edu/~carey/Courses/PSYC5741/handouts/GLM%20Theory.pdf>).

5 - This is not linked to a previous comment of mine, but I noticed that the revised manuscripts states that all data is part of the supplement. However, the only supplementary file I could access did not contain any data but only some additional figures. Please also upload the data file (which, if memory serves me well, was available for the previous manuscript version).

Review form: Reviewer 2

Is the manuscript scientifically sound in its present form?

Yes

Are the interpretations and conclusions justified by the results?

Yes

Is the language acceptable?

Yes

Do you have any ethical concerns with this paper?

No

Have you any concerns about statistical analyses in this paper?

No

Recommendation?

Accept as is

Comments to the Author(s)

The authors have satisfactorily addressed the reviewers' concerns.

Two minor corrections:

Line 121: Burst speed swimming ("speed" missing"

Line 133 Using an electromagnetic flowmeter.

Decision letter (RSOS-201516.R1)

Dear Ms Muhawenimana

On behalf of the Editors, we are pleased to inform you that your Manuscript RSOS-201516.R1 "Temperature surpasses the effects of velocity and turbulence on swimming performance of two invasive non-native fish species" has been accepted for publication in Royal Society Open Science subject to minor revision in accordance with the referees' reports. Please find the referees' comments along with any feedback from the Editors below my signature.

Please submit your revised manuscript and required files (see below) no later than 7 days from today's (ie 21-Jan-2021) date. Note: the ScholarOne system will 'lock' if submission of the revision

is attempted 7 or more days after the deadline. If you do not think you will be able to meet this deadline please contact the editorial office immediately.

on behalf of Pete Smith (Subject Editor)
openscience@royalsociety.org

Associate Editor Comments to Author:

Comments to the Author:

A few minor tweaks remain - we'll look forward to receiving the revision in the near future.

Reviewer comments to Author:

Reviewer: 1

Comments to the Author(s)

I wanted to thank the authors for their careful revisions based on my previous set of comments. I found the paper much improved but also noticed that not all changes had always been done as stated. I therefore list below a few follow-up issues that arose during the revisions.

1 - I had originally asked for clarification on how certain measurements were done (listed as comment 1.2 in the reply to comments). While the authors replied with most of the needed details in the reply document (thank you), they only added some of this to the manuscript. For example, the manuscript still does not state that photographs were taken with an iPhone camera. This needs to be added, as well as which model that iPhone was (since several older models do not really have cameras with a very good resolution, which could impact on the quality of the pictures). Also, they now clarify in the revised manuscript that weight was measured to two decimal places (again, thank you) but still fail to mention what unit of measure was measured to two decimal places. Was this g? Please add this information to the manuscript.

2 - I had mentioned that the font size and format changed between panels within figures. While this was corrected for some (thank you), it was not corrected for all figures. Please also adjust this for all panels in figure 5.

3 - Thank you for clarifying why you only included 'sex' as a factor for one species but not the other. However, does this mean that *Lepomis* were juveniles but *Pseudorasbora* were adults? If

so, then some of the differences you found between species could have easily been also the result of testing juveniles versus adults. This needs to be directly addressed in the discussion.

4 - Thank you for adding the degrees of freedom to the Table 3 header. However, you now list a single degree of freedom for each type of model, is this the error degree of freedom? For different terms in a Gaussian GLM you have to state two different degrees of freedom, the DF associated with that term and the overall error DF for the model (Gotelli & Ellison, 2004, *A Primer of Ecological Statistics*, Sinauer Associates; or see here: <http://psych.colorado.edu/~carey/Courses/PSYC5741/handouts/GLM%20Theory.pdf>).

5 - This is not linked to a previous comment of mine, but I noticed that the revised manuscripts states that all data is part of the supplement. However, the only supplementary file I could access did not contain any data but only some additional figures. Please also upload the data file (which, if memory serves me well, was available for the previous manuscript version).

Reviewer: 2

Comments to the Author(s)

The authors have satisfactorily addressed the reviewers' concerns.

Two minor corrections:

Line 121: Burst speed swimming ("speed" missing"

Line 133 Using an electromagnetic flowmeter.

===PREPARING YOUR MANUSCRIPT===

If you have been asked to revise the written English in your submission as a condition of publication, you must do so, and you are expected to provide evidence that you have received language editing support. The journal would prefer that you use a professional language editing service and provide a certificate of editing, but a signed letter from a colleague who is a native speaker of English is acceptable. Note the journal has arranged a number of discounts for authors

using professional language editing services
(<https://royalsociety.org/journals/authors/benefits/language-editing/>).

===PREPARING YOUR REVISION IN SCHOLARONE===

-- If you have uploaded ESM files, please ensure you follow the guidance at <https://royalsociety.org/journals/authors/author-guidelines/#supplementary-material> to include a suitable title and informative caption. An example of appropriate titling and captioning may be found at https://figshare.com/articles/Table_S2_from_Is_there_a_trade-

off_between_peak_performance_and_performance_breadth_across_temperatures_for_aerobic_sc
ope_in_teleost_fishes_/3843624.

Author's Response to Decision Letter for (RSOS-201516.R1)

See Appendix B.

Decision letter (RSOS-201516.R2)

Dear Ms Muhawenimana,

It is a pleasure to accept your manuscript entitled "Temperature surpasses the effects of velocity and turbulence on swimming performance of two invasive non-native fish species" in its current form for publication in Royal Society Open Science.

Best regards,

on behalf of the Associate Editor and Professor Pete Smith (Subject Editor)
openscience@royalsociety.org

Appendix A

Royal Society Open Science - Manuscript ID RSOS-201516

Temperature surpasses the effects of velocity and turbulence on swimming performance of two invasive non-native fish species

05th December 2020

Dear Royal Society Open Science Editor,

Thank you and the reviewers for the revision of our paper. Below we have outlined our response (in regular font) to the reviewer's comments and suggestions (**in bold font**) and have revised our manuscript accordingly. Line numbers refer to the manuscript with "track changes".

Thank you for your consideration, and we look forward to hearing from you.

Yours sincerely,

Dr Valentine Muhawenimana (on behalf of all authors)

Associate Editor Comments to Author:

Please carefully revise your paper in response to feedback you have received - a revision will be sent to at least one of the reviewers for further oversight, and you should be aware that the journal does not routinely permit multiple rounds of revision.

Reviewer comments to Author:

1. Reviewer: 1

Comments to the Author(s)

In the present study, the authors investigated the effects of temperature, velocity and turbulence on swimming performance of two fishes, invasive to the UK (pumpkinseed and topmouth gudgeon). They tested swimming performance in a flume at three different set temperature while varying velocity within each setting (which also led to changes in turbulence in different parts of the experimental arena). They found that compared to the effects on temperature, effects of velocity and turbulence were relatively small. Specifically, swimming performance largely increased with increasing temperatures. They then discuss these results also with respect to how the invasiveness of these fish might be exacerbated by increasing temperatures resulting from global climate change.

I found this manuscript well written and the methodology appropriate. I did not find any major issues with the manuscript, study design, analysis or interpretation but discovered a few smaller issues that will need to be corrected as part of a revision.

Minor comments:

1.1. A streamlining of the usage of non-native and invasive throughout would be beneficial. Currently, this seems to jump around a bit. For example, in lines 68-69, the pumpkin seed is labelled as non-native while the topmouth gudgeon is labelled as invasive? Aren't they both invasive to the UK?

Thank you, we have used the term “invasive non-native species” throughout the manuscript. Lines 68-69, now lines 69-70 has been amended to “The present study focussed on two invasive non-native UK freshwater fish species, the pumpkinseed (*Lepomis gibbosus*) and the topmouth gudgeon (*Pseudorasbora parva*).”

1.2. Methods, lines 93-94: Regarding the measurements of standard length and weight - What was the measurement level? Was SL quantified to the nearest mm, or even to some nearest decimal of mm? What about weight? Similarly, regarding the mention of a photograph having been taken, please specify what kind of camera this was.

The standard length was measured to the nearest mm and the weight to two decimal places. The photographs were taken using an iPhone camera. We have added these details in lines 95-96.

1.3. Methods, lines 110-112: This needs a bit more information as the numbers in this sentence do not add up. If the inner arena was only 10cm high, then the water depth could not have been 12cm. I am assuming, from looking at Figure 1, that the outer container was much higher than the inner container, so that the water level surpassed

the height of the inner arena? This needs to be clarified as I am only guessing this.

Thank you for pointing this out. The depth of the inner arena was 12 cm, and the water depth was 10 cm. We have corrected this in line 117.

1.4. Results, lines 179-182: This wording here is a bit confusing, as the posthoc tests listed in Table 3 do not always show that the differences that are mentioned here in this part of the text were actually significant. It would be good to also indicate somewhere here in the text which pairwise temp comparisons were actually significant and which were not.

Thank you, we amended this text, now in lines 205-211.

1.5. Figure 3: This could just be an artifact of the file upload, but currently this figure looks like it is relatively low resolution compared to the other figures. If true, please increase resolution. Also, the font format changes between both panels (also in some of the later figures 4, 5 and 6). Please adjust to the same format in both/all panels.

Figure 3 has been corrected as shown below. The formatting and resolution of Figures 4, 5 and 6 have been improved, and we will ensure their resolution is maintained during file uploading.

Figure 3. (A) Burst swimming speed BS (cm s^{-1}) and (B) Time to fatigue T_f (min) of *Lepomis gibbosus* (filled boxes) and *Pseudorasbora parva* (open boxes) at 15, 20 and 25°C. Boxplots indicate, from bottom to top, minimum (lower whisker), first quartile, median, third quartile, maximum (upper whisker), and outliers (circle).

Specific comments:

1.6. Abstract, line 24: Please switch 'on as' to 'as on'

Amended.

1.7. Introduction, lines 35-36: Some conjunctive adverb or similar would be good here, given that this sentence appears to be meant as a single-sentence statement rather than the lead in to an in-depth paragraph of climate change. I suggest something along the lines of: Climate change, on the other hand, impacts all environments, significantly...

Amended.

1.8. Introduction, line 69: Here and throughout - please do not abbreviate the genus name at the beginning of a sentence.

The genus name abbreviations have been corrected throughout the manuscript.

1.9. Introduction, line 72: Please change to 'is subject to'.

Amended, thank you.

1.10. Methods, line 163: Why was 'sex' only included in the models for *P. parva*? I assume this was because *L. gibbosus* lacks sexual dimorphism? Please explain and justify.

We have added in lines 97-100 that pumpkinseed used in the study were immature (confirmed by dissection and examination of gonads) and could not be sexed morphologically. Topmouth gudgeon were sexed following dissection at the end of the study.

1.11. Results, line 173: I am not a native speaker, but this reads odd to me. Maybe rephrase to: '...were on average 29% faster at 20°C but 5% slower at 25°C, than at 15°C.'

Amended.

1.12. Discussion, lines 255-256: There are a lot of 'ands' in this sentence which makes it awkward to read; please rephrase.

Amended.

1.13. References

- **References, line 341: Please remove '(80-).'** from the entry.
- **References, lines 345-347: Please provide journal name, volume and page range.**
- **References, line 378: Please put '*Galaxias maculatus*' in italics.**
- **References, lines 441-442: Please provide page range or e-article number, and please put volume number in bold.**
- **References, line 458: Please delete 'LP' from the middle of the page range.**
- **References, lines 470-471: Please correct the title of this publication, something went wrong there. Also, the page range provided is not correct.**

Thank you for pointing these out, they all have been amended in the reference list.

1.14. Table 3, lines 489-491: What were the degrees of freedom for these models?

We have added the degrees of freedom in Table 3 caption (lines 526).

1.15. Figure 1 legend, line 495: Please change 'no to scale' to 'not to scale'.

Thank you, this has been amended.

2. Reviewer: 2

Comments to the Author(s)

This is a nicely designed study investigating the effects of water temperature, flow velocity and turbulence on swimming performance in two invasive fish species. The strength of this study is the detail of the design and its successful implementation to capture some intricate measurements. The authors were able to quantify swimming performance and behaviour (area of water column occupied) and showed an interplay with thermal conditions and hydrodynamic flow characteristics. I think this work is a valuable addition to the literature and clearly demonstrates better modelling of real-world conditions provides better understanding. The whole manuscript is written in a comprehensive and engaging manner and Table 3 is an excellent way of presenting the statistical results. I have a few comments:

2.1. Please listen to your statistics – temperature had no effect on burst speed swimming in topmouth gudgeon, *Pseudorasbora parva*, yet in the results, discussion and abstract the authors continually state that burst speed swimming was positively affected by temperature for both species. This is not true and the text should be rewritten to make this clear. You have already correctly referred to these results in lines 170-171, but please remove the text “ *P. parva* burst speeds were on average 29% faster but 5% slower at 20 and 25°C, respectively, than at 15°C (Fig. 3B)” note- this is actually Fig. 3A. Please also amend the rest of the manuscript to better reflect your results.

It is correct that the burst swimming of *P. parva* was not overall significantly affected by temperature. We have amended the text to ensure that the text is consistent with the statistics in the results (lines 205-211) and throughout the manuscript. Thank you for pointing this out.

2.2. Continuing the theme that *Lepomis gibbosus* and *P. parva* showed different results, I would like to suggest the different ecologies of the two species should be considered in your interpretation of the results. *P. parvus* is a temperate species (FishBase provides a temperature range of 5 - 22°C) whilst *L. gibbosus* is subtropical (4 - 30°C). The varying effects of different temperatures on burst speed in *P. parvus* may be explained by the upper temperature tested (25°C) exceeding a pejus temperature for this species.

Thank you. We have now included the consideration of different temperature throughout the discussion and specifically in lines 323-325 to highlight that the varied response by the two species likely depend on their native temperature range/physiology.

2.3. The figures are not particularly intuitive and could be simplified to convey the key information. Figure 2 takes a few reads to fully understand it – what does the triangle with three lines underneath it represent? The data are complex, so difficult to visualise but I do feel they could be displayed better in places, especially in Figure 4. In addition, the legend for Figure 5 is incomplete (I am guessing the red squares are 25°C data) and these graphs might work better as regressions plus confidence intervals, rather than showing all the data. Please also continue your convention of using the species name (*L. Gibbosus* and *P. parva*), not the common name, in the figure legend.

The underlined triangle in figure 2 (i) indicates the water surface, and this has been added in the figure caption (line 538). We have also used the genus name throughout the manuscript. The figure 5 legend has been corrected to show all three temperatures.

2.4. There is some inconsistency in use of symbols: l or L for litre (e.g. lines 124 and 128).

Thank you, this has been amended (lines 132 and 134).

3. Reviewer: 3

Comments to the Author(s)

I have reviewed the manuscript RSOS-201516 Temperature surpasses the effects of velocity and turbulence on swimming performance of two invasive non-native fish species by Muhawenimana et al. The manuscript characterises the swimming performances of two freshwater fish species in response to environmental temperature. The authors posit that the performance characteristics of the two species, which are non-native in the UK, could facilitate range expansions with climate change. The manuscript is well written,

however there are several problems with the experimental design and reporting of methodology that prevent me from recommending it for publication at this time.

3.1. There are no ecological data provided on the environments currently inhabited by the two species in the study/sampling location. Thermal data are particularly important in this respect. Do the test temperatures selected represent realistic thermal regimes experienced in their current locations and how well do they reflect proposed thermal regimes under the various IPCC warming scenarios? The findings should be put into more of an ecological context.

In the UK, average high temperatures range between around 15 and 20°C for six months of the year and reach summer highs of 32°C. The IPCC warming scenarios project 0.5°C global warming in the next two to three decades alone (IPCC 2018). The temperatures used in the current study 15, 20 and 25°C, were chosen with these considerations. We also discuss the implications of warmer waters for the fish species evaluated in this study in the introduction and discussion sections.

3.2. There is very little coverage of the thermal physiology of swimming in fish, and more importantly, how different species may deal with thermal variability in their environments. This experimental design only allows for the thermal sensitivity of the performance trait over the thermal range to be examined. There is no consideration of the potential for thermal plasticity (acclimation/acclimatization), adaptation or behavioural changes to offset, or possibly, amplify the effects of climate warming on fish performance.

We appreciate the reviewer's valuable comment; evaluating the thermal physiology and plasticity of these swimming fish is beyond the scope of the current study, but we have now commented on this point (lines 323-325). The aim of this paper is to understand how temperature impacts pumpkinseed and topmouth gudgeon swimming behaviour and evaluate the implication for their dispersal and establishment in the UK, which this paper accomplishes.

3.3. The experiment attempts to link behavioural attributes ('preferences') with turbulence and flow velocity and swimming performance. Firstly, all these experiments were conducted in a long glass swim channel (I think, however details about flume are lacking) with flow straighteners all of which are designed to minimise turbulence in the fluid flow stream. Although I understand that there is some turbulence in the swim channel particularly at the channel walls where the fluid interacts with the structure and this increases with fluid velocity, this level of turbulence would be miniscule compared to the turbulence in natural environments. Velocity and turbulence are highly correlated in this experiment as well. More importantly, the authors have not set out to empirically test how hydrodynamics affects swimming performance and instead are relying on correlations between fish presence and turbulence/velocity to inform their position. I'm not sure that this is a valid or particularly useful way to test this idea.

We have added details of the flume used in the sustained swimming tests (lines 130-135). Swimming tests conducted in swim tunnels are different from those conducted in open channels. Open channel flumes provide a free surface flow which is three-dimensional and thus reproduces more closely the flow in a river than that of a wall-bounded swim channel that resembles pipe flow. The flows examined in this paper are in the fully turbulent regime (Reynolds numbers of 6,600 to 48,780), emulating velocity and turbulent conditions found in a river and the relative importance of fluid friction (viscosity) and flow inertia. Flow straighteners in flumes are used at the upstream inlet to break up the incoming flow pattern and force it into parallel lines as it enters the flume, therefore dampening the incoming turbulence

delivered from a pipe to the upstream flume inlet. The presence of the bed boundary and free water surface at the Reynolds numbers investigated, produces a variation in velocity throughout the water column, resembling the free surface flows found in rivers rather than the uniform distribution of velocity found in pipe flow. Therefore, measuring the swimming performance of the fish in the flume without consideration of the local velocities and turbulence levels they encounter, would be incomplete and neglect what has been shown to affect fish behaviour. Figure 4 and supplementary material (Figures S2 and S3) show the variations of local velocities in the test section. In comparison to natural channels or more hydraulically rough conditions, the turbulence levels in the current experiment are minimal, yet still consequential. However, we recognise that conducting these experiments with more elevated turbulence levels might further this research area.

3.4. There are no hypotheses presented. I do not believe that the gap in knowledge that the MS is attempting to address has been clearly articulated.

The research gaps addressed are given in the introduction (lines 62-68 and lines 77-79) and the aim of the paper is given in lines 79-82.

3.5. Statistics and methods need greater attention. There are insufficient details provided about the flume and its operation, there are no details provided on the generation of the thermal treatments for the sustained swimming tests, there are no details about how behavioural data were collected from videos and how these may have been analysed statistically. How long were fish given to adjust to thermal conditions before testing? Were they fasted before testing? What time of day was testing conducted? Were fish re-swum for any tests or were unique individuals used for each metric, if so were repeated measures tests considered? Turbulence and velocity metrics are likely to all be highly correlated, so the statistical treatment of these variables needs further consideration (multicollinearity). I would like to see the statistical models more fully detailed, particularly how the hydrodynamic variables were considered as they related to performance/behaviour.

We have improved the methods sections by providing further details of the flume used in the sustained swimming tests (lines 130-135). Fish were given 1 month to adjust to the test temperatures (lines 106-107). We added fish handling details in section 2.1 (lines 84-109): e.g. fish were fasted for 12 hours prior to testing, which took place during daytime between 8:00 and 18:00h; and individual fish were tested once per treatment in lines 107, 109, 116) Information about the sustained swimming test have been added in lines 137-138, 144, 148-149, 153). We also added details of statistical analysis in lines 183-187.

3.6. There is a disjuncture between the premise for the manuscript established in the introduction and the conclusions drawn in the discussion. The introduction sets the MS up to be primarily about invasive species management and projected range expansions with climate change, while the discussion is dominated largely by consideration of swimming position in the flume and effects of body shape and temperature thereon. While there is a place for the authors to consider the implications of their research for invasive species management, the design of the study does not really support this being the dominant focus of the manuscript, and so the introduction needs to be suitably revised. The study is primarily about the thermal sensitivity of performance (and to a lesser extent, behaviour) in these two fish species. If the authors want to take the invasive species angle, then they should be to be comparing performance and behaviour across native and other non-native species (in a phylogenetically considered manner).

We have slightly reduced the emphasis on invasive non-native species, but as we tested several environmental and hydrodynamic factors that could influence the swimming performance of invasive non-native species (changes in performance due to temperature, local velocities and turbulence and differences between species) we do think this is a valid angle; particularly because a driver for this work was the Environment Agency wanting to know the answers to these questions in relation to their control of invasive non-native species.

Minor points:

3.7. line 39: reference required after ‘drivers’.

A reference [8] (Walther et al. 2009) has been added (line 39)

3.8. Line 62-67: The authors need to broaden their literature search criteria or reduce the generality of these statements. There are countless flume studies looking at the effects of temperature on fish swimming performance – the majority of these manipulate fluid flow velocity to measure swimming performance.

We have amended this sentence (line 66) to focus on the studies that evaluate flow velocity and turbulence effects on swimming performance. While we agree that there is a broad body of literature evaluating fish swimming performance in relationship to velocity and temperature, they are conducted in swimming tunnels, which are designed to have homogenous velocity distributions; in contrast to open channels which present varied local velocities, therefore impacting swimming performance (please refer to our response to point 3.3. above).

3.9. In 68: insert ‘freshwater fish’ before ‘species

Thank you, this has been amended (line 69).

3.10. Ln 113: how did you deal with the three measures? Averaged? Fastest?

The C-start measure used averaged speed. We have added this in line 120-122.

3.11. Ln 116: filming speed (frame grab rate)? Describe how fish movement was actually measured. Did you only use recordings where you got an actual ‘C’ start? Report values in SI units (m s⁻¹).

The startle responses were recorded using a using a digital camera (Sony HDR-CX405) with 1440x1080 resolution at 60 frames per second, which we have added in lines 124-125.

3.12. Section 2.3 – please provide a more detailed description of flume design (construction material, method for generation of fluid movement, baffling, header tanks, downstream gates etc.). Description of generation of thermal treatments is lacking in this section. How were fish introduced into the flume? How did you define fatigue? Please describe how behaviour metrics were analysed (define each of the behaviour metrics).

In section 2.3, information about the flume, and water heating and cooling details have now been added (lines 130-135). Lines 144, 148-149, and 153 add information about fish test protocol and recorded behaviour.

3.13. Line 145: grammar

Thank you, this sentence has been corrected. “This made six profiles with six points per Reynolds number.” (line 164).

3.14. line 156: what is ρ ? Were measurements repeated at each velocity increment and at each temperature.

ρ is the density of water, which we have added the text (line 176-177). Velocity measurements were conducted for each velocity increment (line 164) and Table 2 caption (line 515) at 15°C

condition. Temperature does not significantly affect water flow velocity in open channels, and therefore ADV measurements were not repeated for each temperature.

3.15. Ln 163 – 166: temperature? Please define all model response variables and independent variables. Post-hoc tests? What packages/functions were used? Citations for these?

We have added further details of the statistical analysis in lines 183-187.

3.16. Section 3.3. There is so much behavioural methodology missing that don't feel that I can appropriately assess this section.

We have amended the methods section to provide more details of the fish behaviour tests methodology, (lines 130-135, 1424, 148-149, and 153).

3.17. Ln 263-268 – these are a repetition of the results

This paragraph aims to provide a brief summary of the results to transition into the discussion.

3.18. In 285: what is meant by size effects?

Size effects referred to the positive relationship between fish size and sustained swimming performance. We agree though that this sentence in the discussion is confusing and have now omitted this.

3.19. Table 2 – you don't need the te and ts columns. It's enough to say that the time increment at each velocity was 10 min.

Agree, although we think it adds to clarity. However, to reduce space we have now combined this information into a single column.

3.20. In 485 what was the value for water kinematic velocity?

The water kinematic viscosity (ν) depends on the temperature and is 1.14×10^{-6} , 1×10^{-6} and $0.96 \times 10^{-6} \text{ m}^2\text{s}^{-1}$ for 15, 20 and 25°C respectively We have added this information in the table caption (line 519-520) and thank the reviewer for pointing it out.

3.21. Line 506 & 520: Please provide the definitions for the abbreviations in these figure legends. Legends should be able to stand on their own without reference to the main body text.

Thank you. We have amended the figure captions to define the abbreviated terms in lines 546-548 and line 560.

References

- IPCC, 2018: Summary for Policymakers. In: Global warming of 1.5°C. An IPCC Special Report on the impacts of global warming of 1.5°C above pre-industrial levels and related global greenhouse gas emission pathways, in the context of strengthening the global response to the threat of climate change, sustainable development, and efforts to eradicate poverty (V. Masson-Delmotte, P. Zhai, H. O. Pörtner, et al. (eds.)). World Meteorological Organization, Geneva, Switzerland, 32 pp.
- Walther G-R et al. 2009 Alien species in a warmer world: risks and opportunities. *Trends Ecol. Evol.* 24, 686–693. (doi:10.1016/j.tree.2009.06.008)

Appendix B

Royal Society Open Science - Manuscript ID RSOS-201516.R1

Temperature surpasses the effects of velocity and turbulence on swimming performance of two invasive non-native fish species

25th January 2021

Dear Royal Society Open Science Editor,

Thank you for revising and accepting our manuscript. We have addressed the minor comments from the reviewers (**in bold font**), and outline below the changes made (in regular font). Line numbers refer to the text with tracked changes.

Thank you for your consideration, and we look forward to hearing from you.

Yours sincerely,

Dr Valentine Muhawenimana (on behalf of all authors)

Associate Editor Comments to Author:

Comments to the Author:

A few minor tweaks remain - we'll look forward to receiving the revision in the near future.

Reviewer comments to Author:

Reviewer: 1

Comments to the Author(s)

I wanted to thank the authors for their careful revisions based on my previous set of comments. I found the paper much improved but also noticed that not all changes had always been done as stated. I therefore list below a few follow-up issues that arose during the revisions.

1 - I had originally asked for clarification on how certain measurements were done (listed as comment 1.2 in the reply to comments). While the authors replied with most of the needed details in the reply document (thank you), they only added some of this to the manuscript. For example, the manuscript still does not state that photographs were taken with an iPhone camera. This needs to be added, as well as which model that iPhone was (since several older models do not really have cameras with a very good resolution, which could impact on the quality of the pictures). Also, they now clarify in the revised manuscript that weight was measured to two decimal places (again, thank you) but still fail to mention what unit of measure was measured to two decimal places. Was this g? Please add this information to the manuscript.

Apologies and thank you for pointing this out. We have added in line 97 that the model used was iPhone 6S, and the weight measurement unit g in line 96.

2 - I had mentioned that the font size and format changed between panels within figures. While this was corrected for some (thank you), it was not corrected for all figures. Please also adjust this for all panels in figure 5.

Thank you, the format has now been amended in figure 5.

3 - Thank you for clarifying why you only included 'sex' as a factor for one species but not the other. However, does this mean that Lepomis were juveniles but Pseudorasbora were adults? If so, then some of the differences you found between species could have easily been also the result of testing juveniles versus adults. This needs to be directly addressed in the discussion.

In the discussion section, line 316 has been amended to take into consideration that the variations in behaviour could be attributed to the different ages of the fish. Thank you.

4 - Thank you for adding the degrees of freedom to the Table 3 header. However, you now list a single degree of freedom for each type of model, is this the error degree of freedom? For different terms in a Gaussian GLM you have to state two different degrees of freedom, the DF associated with that term and the overall error DF for the model (Gotelli & Ellison, 2004, A Primer of Ecological Statistics, Sinauer Associates; or see here:

<http://psych.colorado.edu/~carey/Courses/PSYC5741/handouts/GLM%20Theory.pdf>).

We have clarified in Table 3 caption (line 516) that the degrees of freedom are associated with null and residual deviance, and thank you for the linked document.

5 - This is not linked to a previous comment of mine, but I noticed that the revised manuscripts states that all data is part of the supplement. However, the only supplementary file I could access did not contain any data but only some additional figures. Please also upload the data file (which, if memory serves me well, was available for the previous manuscript version).

The Excel data file entitled “Dataset_L gibbosus and P parva” contains the data associated with the manuscript, and the “Supplementary material” Word document contains supplementary figures. We have ensured that both are uploaded in the submission.

Reviewer: 2

Comments to the Author(s)

The authors have satisfactorily addressed the reviewers' concerns.

Two minor corrections:

Line 121: Burst speed swimming ("speed" missing"

Thank you, this is now amended.

Line 133 Using an electromagnetic flowmeter.

Amended, thank you.